# Partial success in closing the gap between human and machine vision

**Robert Geirhos**[1-2][§]    **Kantharaju Narayanappa**[1]    **Benjamin Mitzkus**[1]

**Tizian Thieringer**[1]    **Matthias Bethge**[1][*]    **Felix A. Wichmann**[1][*]    **Wieland Brendel**[1][*]

[1]University of Tübingen
[2]International Max Planck Research School for Intelligent Systems
[*]Joint senior authors
[§]To whom correspondence should be addressed: `robert.geirhos@uni-tuebingen.de`

## Abstract

A few years ago, the first CNN surpassed human performance on ImageNet. However, it soon became clear that machines lack robustness on more challenging test cases, a major obstacle towards deploying machines "in the wild" and towards obtaining better computational models of human visual perception. Here we ask: Are we making progress in closing the gap between human and machine vision? To answer this question, we tested human observers on a broad range of out-of-distribution (OOD) datasets, recording 85,120 psychophysical trials across 90 participants. We then investigated a range of promising machine learning developments that crucially deviate from standard supervised CNNs along three axes: objective function (self-supervised, adversarially trained, CLIP language-image training), architecture (e.g. vision transformers), and dataset size (ranging from 1M to 1B).

Our findings are threefold. (1.) The longstanding *distortion robustness gap* between humans and CNNs is closing, with the best models now exceeding human feedforward performance on most of the investigated OOD datasets. (2.) There is still a substantial image-level *consistency gap*, meaning that humans make different errors than models. In contrast, most models systematically agree in their categorisation errors, even substantially different ones like contrastive self-supervised vs. standard supervised models. (3.) In many cases, human-to-model consistency improves when training dataset size is increased by one to three orders of magnitude. Our results give reason for cautious optimism: While there is still much room for improvement, the behavioural difference between human and machine vision is narrowing. In order to measure future progress, 17 OOD datasets with image-level human behavioural data and evaluation code are provided as a toolbox and benchmark at `https://github.com/bethgelab/model-vs-human/`.

## 1   Introduction

Looking back at the last decade, deep learning has made tremendous leaps of progress by any standard. What started in 2012 with AlexNet [1] as the surprise winner of the ImageNet Large-Scale Visual Recognition Challenge quickly became the birth of a new AI "summer", a summer lasting much longer than just a season. With it, just like with any summer, came great expectations: the hope that the deep learning revolution will see widespread applications in industry, that it will propel breakthroughs in the sciences, and that it will ultimately close the gap between human and machine

35th Conference on Neural Information Processing Systems (NeurIPS 2021).

perception. We have now reached the point where deep learning has indeed become a significant driver of progress in industry [e.g. 2, 3], and where many disciplines are employing deep learning for scientific discoveries [4–9]—*but are we making progress in closing the gap between human and machine vision?*

**IID vs. OOD benchmarking.**    For a long time, the gap between human and machine vision was mainly approximated by comparing benchmark accuracies on IID (independent and identically distributed) test data: as long as models are far from reaching human-level performance on challenging datasets like ImageNet, this approach is adequate [10]. Currently, models are routinely matching and in many cases even outperforming humans on IID data. At the same time, it is becoming increasingly clear that models systematically exploit shortcuts shared between training and test data [11–14]. Therefore we are witnessing a major shift towards measuring model performance on out-of-distribution (OOD) data rather than IID data alone, which aims at testing models on more challenging test cases where there is still a ground truth category, but certain image statistics differ from the training distribution. Many OOD generalisation tests have been proposed: ImageNet-C [15] for corrupted imaes, ImageNet-Sketch [16] for sketches, Stylized-ImageNet [17] for image style changes, [18] for unfamiliar object poses, and many more [19–29]. While it is great to have many viable and valuable options to measure generalisation, most of these datasets unfortunately lack human comparison data. This is less than ideal, since we can no longer assume that humans reach near-ceiling accuracies on these challenging test cases as they do on standard noise-free IID object recognition datasets. In order to address this issue, we carefully tested human observers in the Wichmannlab's vision laboratory on a broad range of OOD datasets, providing some 85K psychophysical trials across 90 participants. Crucially, we showed exactly the same images to multiple observers, which means that we are able to compare human and machine vision on the fine-grained level of individual images [30–32]). The focus of our datasets is measuring *distortion robustness*: we tested 17 variations that include changes to image style, texture, and various forms of synthetic additive noise.

**Contributions & outlook.**    The resulting 17 OOD datasets with large-scale human comparion data enable us to investigate recent exciting machine learning developments that crucially deviate from "vanilla" CNNs along three axes: objective function (supervised vs. self-supervised, adversarially trained, and CLIP's joint language-image training), architecture (convolutional vs. vision transformer) and training dataset size (ranging from 1M to 1B images). Taken together, these are some of the most promising directions our field has developed to date—but this field would not be machine learning if new breakthroughs weren't within reach in the next few weeks, months and years. Therefore, we open-sourced `modelvshuman`, a Python toolbox that enables testing both PyTorch and TensorFlow models on our comprehensive benchmark suite of OOD generalisation data in order to measure future progress. Even today, our results give cause for (cautious) optimism. After a method overview (Section 2), we are able to report that the human-machine *distortion robustness gap* is closing: the best models now match or in many cases even exceed human feedforward performance on most of the investigated OOD datasets (Section 3). While there is still a substantial image-level *consistency gap* between humans and machines, this gap is narrowing on some—but not all—datasets when the size of the training dataset is increased (Section 4).

## 2   Methods: datasets, psychophysical experiments, models, metrics, toolbox

**OOD datasets with consistency-grade human data.**    We collected human data for 17 generalisation datasets (visualized in Figures 7 and 8 in the Appendix, which also state the number of subjects and trials per experiment) on a carefully calibrated screen in a dedicated psychophysical laboratory (a total of 85,120 trials across 90 observers). Five datasets each correspond to a single manipulation (sketches, edge-filtered images, silhouettes, images with a texture-shape cue conflict, and stylized images where the original image texture is replaced by the style of a painting); the remaining twelve datasets correspond to parametric image degradations (e.g. different levels of noise or blur). Those OOD datasets have in common that they are designed to test ImageNet-trained models. OOD images were obtained from different sources: sketches from ImageNet-Sketch [16], stylized images from

Stylized-ImageNet [17], edge-filtered images, silhouettes and cue conflict images from [17][1], and the remaining twelve parametric datasets were adapted from [33]. For these parametric datasets, [33] collected human accuracies but unfortunately, they showed different images to different observers implying that we cannot use their human data to assess image-level consistency between humans and machines. Thus we collected psychophysical data for those images ourselves by showing exactly the same images to multiple observers for each of those twelve datasets. Additionally, we cropped the images from [33] to $224 \times 224$ pixels to allow for a fair comparison to ImageNet models (all models included in our comparison receive $224 \times 224$ input images; [33] showed $256 \times 256$ images to human observers in many cases).

**Psychophysical experiments.** 90 observers were tested in a darkened chamber. Stimuli were presented at the center of a 22" monitor with $1920 \times 1200$ pixels resolution (refresh rate: 120 Hz). Viewing distance was 107 cm and target images subtended $3 \times 3$ degrees of visual angle. Human observers were presented with an image and asked to select the correct category out of 16 basic categories (such as chair, dog, airplane, etc.). Stimuli were balanced w.r.t. classes and presented in random order. For ImageNet-trained models, in order to obtain a choice from the same 16 categories, the 1,000 class decision vector was mapped to those 16 classes using the WordNet hierarchy [34]. In Appendix I, we explain why this mapping is optimal. We closely followed the experimental protocol defined by [33], who presented images for 200 ms followed by a $1/f$ backward mask to limit the influence of recurrent processing (otherwise comparing to feedforward models would be difficult). Further experimental details are provided in Appendix C.

**Why not use crowdsourcing instead?** Our approach of investigating few observers in a high-quality laboratory setting performing many trials is known as the so-called "small-N design", the bread-and-butter approach in high-quality psychophysics—see, e.g., the review "Small is beautiful: In defense of the small-N design" [35]. This is in contrast to the "crowdsourcing approach" (many observers in a noisy setting performing fewer trials each). The highly controlled conditions of the Wichmannlab's psychophysical laboratory come with many advantages over crowdsourced data collection: precise timing control (down to the millisecond), carefully calibrated monitors (especially important for e.g. low-contrast stimuli), controlled viewing distance (important for foveal presentation), full visual acuity (we performed an acuity test with every observer prior to the experiment), observer attention (e.g. no multitasking or children running around during an experiment, which may happen in a crowdsourcing study), just to name a few [36]. Jointly, these factors contribute to high data quality.

**Models.** In order to disentangle the influence of objective function, architecture and training dataset size, we tested a total of 52 models: 24 standard ImageNet-trained CNNs [37], 8 self-supervised models [38–43],[2] 6 Big Transfer models [45], 5 adversarially trained models [46], 5 vision transformers [47, 48], two semi-weakly supervised models [49] as well as Noisy Student [50] and CLIP [51]. Technical details for all models are provided in the Appendix.

**Metrics.** In addition to *OOD accuracy* (averaged across conditions and datasets), the following three metrics quantify how closely machines are aligned with the decision behaviour of humans.

*Accuracy difference $A(m)$* is a simple aggregate measure that compares the accuracy of a machine $m$ to the accuracy of human observers in different out-of-distribution tests,

$$A(m) : \mathbb{R} \to [0, 1], m \mapsto \frac{1}{|D|} \sum_{d \in D} \frac{1}{|H_d|} \sum_{h \in H_d} \frac{1}{|C_d|} \sum_{c \in C_d} (\mathrm{acc}_{d,c}(h) - \mathrm{acc}_{d,c}(m))^2 \qquad (1)$$

where $\mathrm{acc}_{d,c}(\cdot)$ is the accuracy of the model or the human on dataset $d \in D$ and condition $c \in C_d$ (e.g. a particular noise level), and $h \in H_D$ denotes a human observer tested on dataset $d$. Analogously, one can compute the average accuracy difference between a human observer $h_1$ and all other human observers by substituting $h_1$ for $m$ and $h \in H_D \setminus \{h_1\}$ for $h \in H_D$ (which can also be applied for the two metrics defined below).

---

[1]For those three datasets consisting of 160, 160 and 1280 images respectively, consistency-grade psychophysical data was already collected by the authors and included in our benchmark with permission from the authors.

[2]We presented a preliminary and much less comprehensive version of this work at the NeurIPS 2020 workshop SVRHM [44].

Aggregated metrics like $A(m)$ ignore individual image-level decisions. Two models with vastly different image-level decision behaviour might still end up with the same accuracies on each dataset and condition. Hence, we include two additional metrics in our benchmark that are sensitive to decisions on individual images.

*Observed consistency* $O(m)$ [32] measures the fraction of samples for which humans and a model $m$ get the same sample either both right *or* both wrong. More precisely, let $b_{h,m}(s)$ be one if both a human observer $h$ and $m$ decide either correctly or incorrectly on a given sample $s$, and zero otherwise. We calculate the average observed consistency as

$$O(m) : \mathbb{R} \to [0,1], m \mapsto \frac{1}{|D|} \sum_{d \in D} \frac{1}{|H_d|} \sum_{h \in H_d} \frac{1}{|C_d|} \sum_{c \in C_d} \frac{1}{|S_{d,c}|} \sum_{s \in S_{d,c}} b_{h,m}(s) \tag{2}$$

where $s \in S_{d,c}$ denotes a sample $s$ (in our case, an image) of condition $c$ from dataset $d$. Note that this measure can only be zero if the accuracy of $h$ and $m$ are exactly the same in each dataset and condition.

*Error consistency* $E(m)$ [32] tracks whether there is above-chance consistency. This is an important distinction, since e.g. two decision makers with 95% accuracy each will have at least 90% observed consistency, even if their 5% errors occur on non-overlapping subsets of the test data (intuitively, they both get most images correct and thus observed overlap is high). To this end, error consistency (a.k.a. Cohen's kappa, cf. [52]) indicates whether the observed consistency is larger than what could have been expected given two independent binomial decision makers with matched accuracy, which we denote as $\hat{o}_{h,m}$. This can easily be computed analytically [e.g. 32, equation 1]. Then, the average error consistency is given by

$$E(m) : \mathbb{R} \to [-1,1], m \mapsto \frac{1}{|D|} \sum_{d \in D} \frac{1}{|H_d|} \sum_{h \in H_d} \frac{1}{|C_d|} \sum_{c \in C_d} \frac{(\frac{1}{|S_{d,c}|} \sum_{s \in S_{d,c}} b_{h,m}(s)) - \hat{o}_{h,m}(S_{d,c})}{1 - \hat{o}_{h,m}(S_{d,c})} \tag{3}$$

**Benchmark & toolbox.** $A(m)$, $O(m)$ and $E(m)$ each quantify a certain aspect of the human-machine gap. We use the mean rank order across these metrics to determine an overall model ranking (Table 2 in the Appendix). However, we would like to emphasise that the primary purpose of this benchmark is to generate insights, not winners. Since insights are best gained from detailed plots and analyses, we open-source `modelvshuman`, a Python project to benchmark models against human data.[3] The current model zoo already includes 50+ models, and an option to add new ones (both PyTorch and TensorFlow). Evaluating a model produces a 15+ page report on model behaviour. All plots in this paper can be generated for future models—to track whether they narrow the gap towards human vision, or to determine whether an algorithmic modification to a baseline model (e.g., an architectural improvement) changes model behaviour.

## 3   Robustness across models: the OOD distortion robustness gap between human and machine vision is closing

We are interested in measuring whether we are making progress in closing the gap between human and machine vision. For a long time, CNNs were unable to match human robustness in terms of generalisation beyond the training distribution—a large OOD *distortion robustness gap* [14, 33, 53–55]. Having tested human observers on 17 OOD datasets, we are now able to compare the latest developments in machine vision to human perception. Our core results are shown in Figure 1: the OOD distortion robustness gap between human and machine vision is closing (1a, 1b), especially for models trained on large-scale datasets. On the individual image level, a human-machine consistency gap remains (especially 1d), which will be discussed later.

**Self-supervised models**   *"If intelligence is a cake, the bulk of the cake is unsupervised learning, the icing on the cake is supervised learning and the cherry on the cake is reinforcement learning"*,

---

[3]Of course, comparing human and machine vision is not limited to object recognition behaviour: other comparisons may be just as valid and interesting.

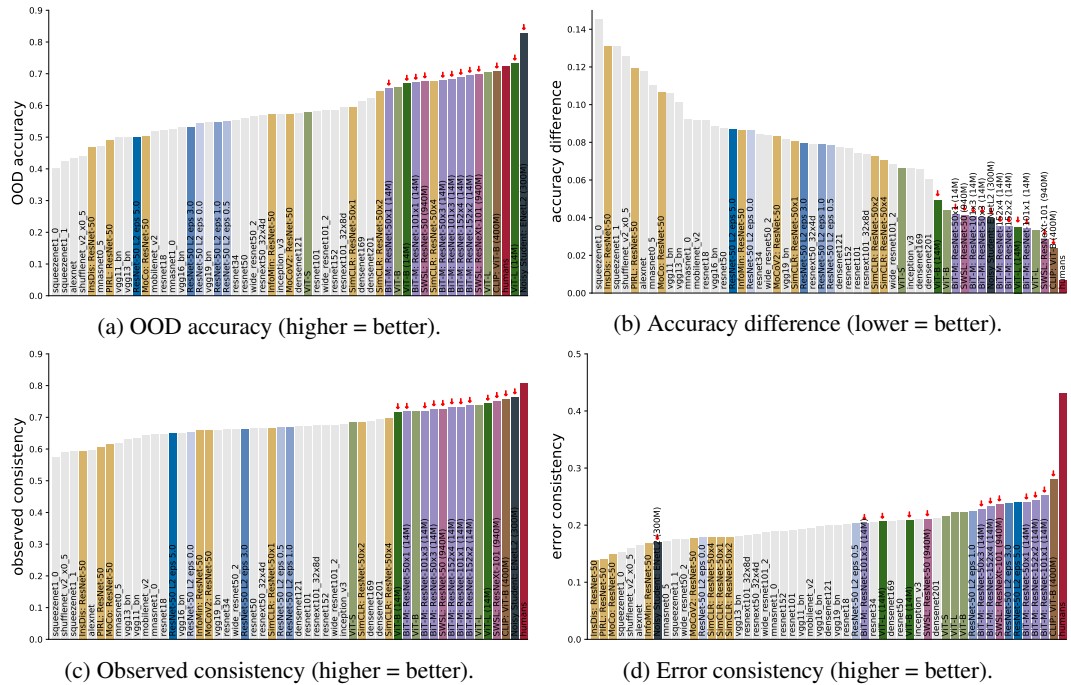

(a) OOD accuracy (higher = better).

(b) Accuracy difference (lower = better).

(c) Observed consistency (higher = better).

(d) Error consistency (higher = better).

Figure 1: Core results, aggregated over 17 out-of-distribution (OOD) datasets: The OOD robustness gap between human and machine vision is closing (top), but an image-level consistency gap remains (bottom). Results compare humans, standard supervised CNNs, self-supervised models, adversarially trained models, vision transformers, noisy student, BiT, SWSL and CLIP. For convenience, ↓ marks models that are trained on large-scale datasets. Metrics defined in Section 2. Best viewed on screen.

Yann LeCun said in 2016 [56]. A few years later, the entire cake is finally on the table—the representations learned via self-supervised learning[4] now compete with supervised methods on ImageNet [43] and outperform supervised pre-training for object detection [41]. But how do recent self-supervised models differ from their supervised counterparts in terms of their behaviour? Do they bring machine vision closer to human vision? Humans, too, rapidly learn to recognise new objects without requiring hundreds of labels per instance; additionally a number of studies reported increased similarities between self-supervised models and human perception [57–61]. Figure 2 compares the generalisation behaviour of eight self-supervised models in orange (PIRL, MoCo, MoCoV2, InfoMin, InsDis, SimCLR-x1, SimCLR-x2, SimCLR-x4)—with 24 standard supervised models (grey). We find only marginal differences between self-supervised and supervised models: Across distortion types, self-supervised networks are well within the range of their poorly generalising supervised counterparts. However, there is one exception: the three SimCLR variants show strong generalisation improvements on uniform noise, low contrast, and high-pass images, where they are the three top-performing self-supervised networks—quite remarkable given that SimCLR models were trained on a different set of augmentations (random crop with flip and resize, colour distortion, and Gaussian blur). Curious by the outstanding performance of SimCLR, we asked whether the self-supervised objective function or the choice of training data augmentations was the defining factor. When comparing self-supervised SimCLR models with augmentation-matched baseline models trained in the standard supervised fashion (Figure 15 in the Appendix), we find that the augmentation scheme (rather than the self-supervised objective) indeed made the crucial difference: supervised baselines show just the same generalisation behaviour, a finding that fits well with [62], who observed that the influence of training data augmentations is stronger than the role of architecture or training objective. In conclusion, our analyses indicate that the "cake" of contrastive self-supervised learning currently (and disappointingly) tastes much like the "icing".

---

[4]"Unsupervised learning" and "self-supervised learning" are sometimes used interchangeably. We use the term "self-supervised learning'" since those methods use (label-free) supervision.

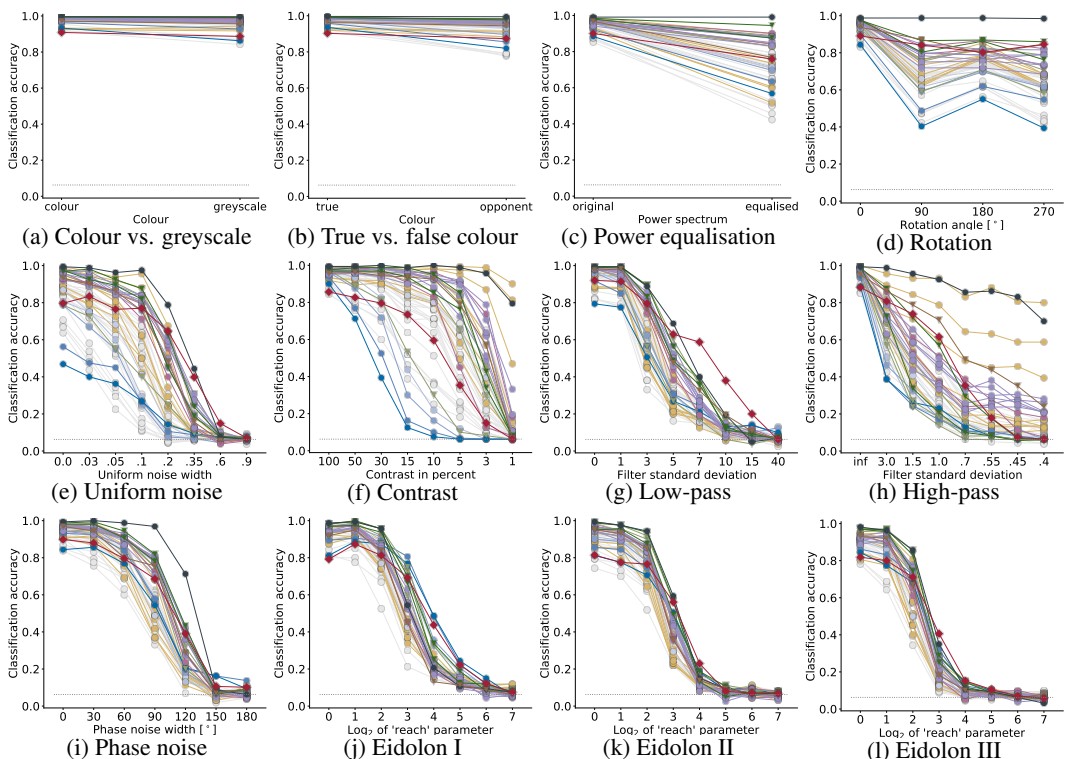

Figure 2: The OOD distortion robustness gap between human and machine vision is closing. Robustness towards parametric distortions for humans, standard supervised CNNs, self-supervised models, adversarially trained models, vision transformers, noisy student, BiT, SWSL, CLIP. Symbols indicate architecture type (○ convolutional, ▽ vision transformer, ◊ human); best viewed on screen.

**Adversarially trained models** The vulnerability of CNNs to adversarial input perturbations is, arguably, one of the most striking shortcomings of this model class compared to robust human perception. A successful method to increase adversarial robustness is *adversarial training* [e.g. 63, 64]. The resulting models were found to transfer better, have meaningful gradients [65], and enable interpolating between two input images [66]: "robust optimization can actually be viewed as inducing a *human prior* over the features that models are able to learn" [67, p. 10]. Therefore, we include five models with a ResNet-50 architecture and different accuracy-robustness tradeoffs, adversarially trained on ImageNet with Microsoft-scale resources by [46] to test whether models with "perceptually-aligned representations" also show human-aligned OOD generalisation behaviour—as we would hope. This is not the case: the stronger the model is trained adversarially (darker shades of blue in Figure 2), the more susceptible it becomes to (random) image degradations. Most strikingly, a simple rotation by 90 degrees leads to a 50% drop in classification accuracy. Adversarial robustness seems to come at the cost of increased vulnerability to large-scale perturbations.[5] On the other hand, there is a silver lining: when testing whether models are biased towards texture or shape by testing them on cue conflict images (Figure 3), in accordance with [69, 70] we observe a perfect relationship between shape bias and the degree of adversarial training, a big step in the direction of human shape bias (and a stronger shape bias than nearly all other models).

**Vision transformers** In computer vision, convolutional networks have become by far the dominating model class over the last decade. Vision transformers [47] break with the long tradition of using convolutions and are rapidly gaining traction [71]. We find that the best vision transformer (ViT-L trained on 14M images) even *exceeds* human OOD accuracy (Figure 1a shows the average across 17 datasets). There appears to be an additive effect of architecture and data: vision transformers trained on 1M images (light green) are already better than standard convolutional models; training on 14M images (dark green) gives another performance boost. In line with [72, 73], we observe a higher shape bias compared to most standard CNNs.

---

[5]This might be related to [68], who studied a potentially related tradeoff between selectivity and invariance.

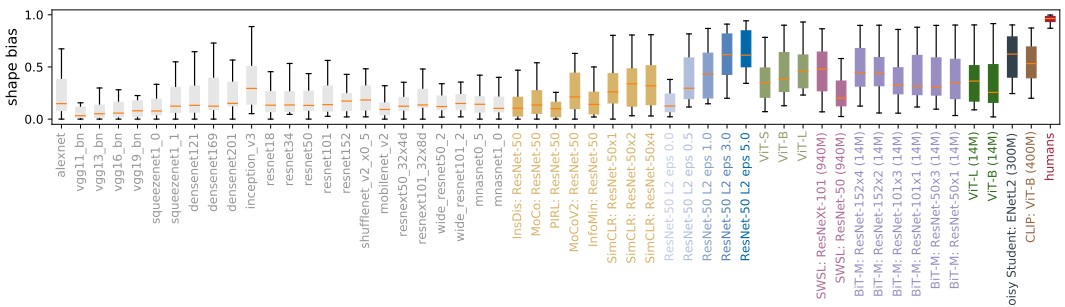

Figure 3: Shape vs. texture biases of different models. While human shape bias is not yet matched, several approaches improve over vanilla CNNs. Box plots show category-dependent distribution of shape / texture biases (shape bias: high values, texture bias: low values).

**Standard models trained on more data: BiT-M, SWSL, Noisy Student** Interestingly, the biggest effect on OOD robustness we find simply comes from training on larger datasets, not from advanced architectures. When standard models are combined with large-scale training (14M images for BiT-M, 300M for Noisy Student and a remarkable 940M for SWSL), OOD accuracies reach levels not known from standard ImageNet-trained models; these models even outperform a more powerful architecture (vision transformer ViT-S) trained on less data (1M) as shown in Figure 1a. Simply training on (substantially) more data substantially narrows the gap to human OOD accuracies (1b), a finding that we quantified in Appendix H by means of a regression model. (The regression model also revealed a significant interaction between dataset size and objective function, as well as a significant main effect for transformers over CNNs.) Noisy Student in particular outperforms humans by a large margin overall (Figure 1a)—the beginning of a new human-machine gap, this time in favour of machines?

**CLIP** CLIP is special: trained on 400M images[6] (more data) with joint language-image supervision (novel objective) and a vision transformer backbone (non-standard architecture), it scores close to humans across all of our metrics presented in Figure 1; most strikingly in terms of error consistency (which will be discussed in the next section). We tested a number of hypotheses to disentangle why CLIP appears "special". *H1: because CLIP is trained on a lot of data?* Presumably no: Noisy Student—a model trained on a comparably large dataset of 300M images—performs very well on OOD accuracy, but poorly on error consistency. A caveat in this comparison is the quality of the labels: while Noisy Student uses pseudolabeling, CLIP receives web-based labels for all images. *H2: because CLIP receives higher-quality labels?* About 6% of ImageNet labels are plainly wrong [74]. Could it be the case that CLIP simply performs better since it doesn't suffer from this issue? In order to test this, we used CLIP to generate new labels for all 1.3M ImageNet images: (a) hard labels, i.e. the top-1 class predicted by CLIP; and (b) soft labels, i.e. using CLIP's full posterior distribution as a target. We then trained ResNet-50 from scratch on CLIP hard and soft labels (for details see Appendix E). However, this does not show any robustness improvements over a vanilla ImageNet-trained ResNet-50, thus different/better labels are not a likely root cause. *H3: because CLIP has a special image+text loss?* Yes and no: CLIP training on ResNet-50 leads to astonishingly poor OOD results, so training a standard model with CLIP loss alone is insufficient. However, while neither architecture nor loss alone sufficiently explain why CLIP is special, we find a clear interaction between architecture and loss (described in more detail in the Appendix along with the other "CLIP ablation" experiments mentioned above).

## 4 Consistency between models: data-rich models narrow the substantial image-level consistency gap between human and machine vision

In the previous section we have seen that while self-supervised and adversarially trained models lack OOD distortion robustness, models based on vision transformers and/or trained on large-scale datasets now match or exceed human feedforward performance on most datasets. Behaviourally, a

---

[6]The boundary between IID and OOD data is blurry for networks trained on big proprietary datasets. We consider it unlikely that CLIP was exposed to many of the exact distortions used here (e.g. eidolon or cue conflict images), but CLIP likely had greater exposure to some conditions such as grayscale or low-contrast images.

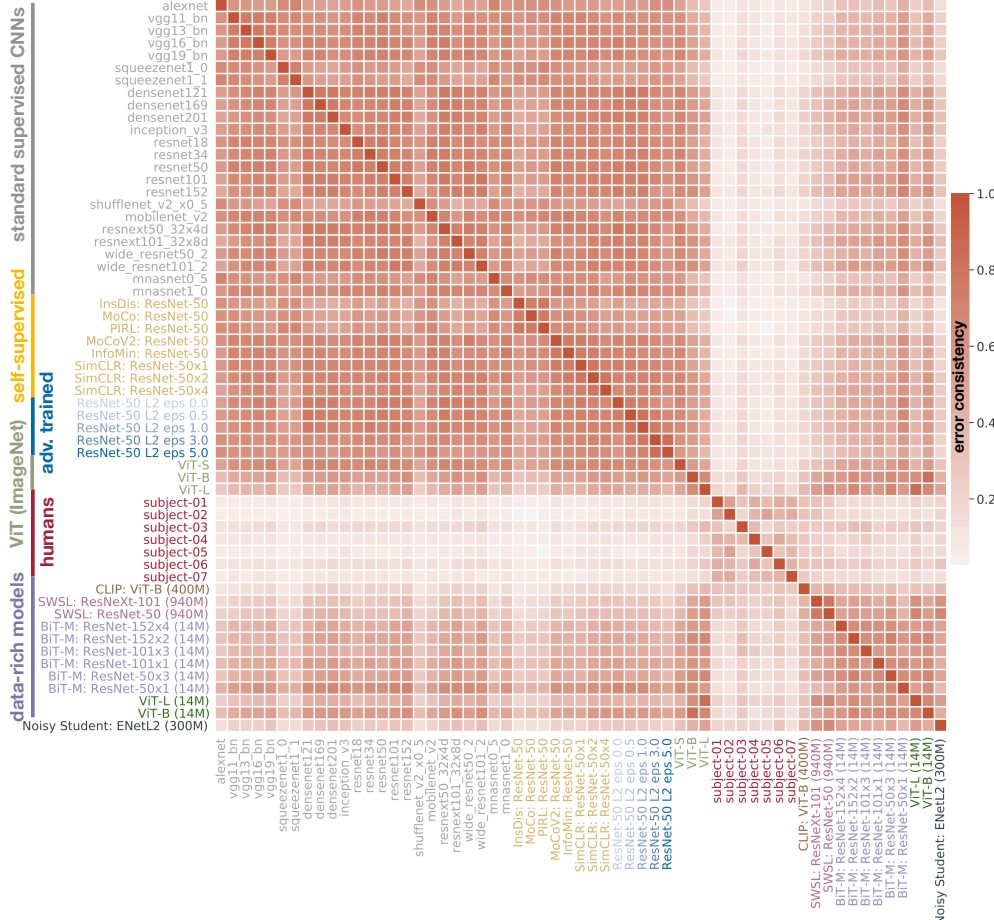

Figure 4: Data-rich models narrow the substantial image-level consistency gap between humans and machines. Error consistency analysis on a single dataset (sketch images; for other datasets see Appendix, Figures 9, 11, 12, 13, 14) shows that most models cluster (dark red = highly consistent errors) irrespective of their architecture and objective function; humans cluster differently (high human-to-human consistency, low human-to-model consistency); but some data-rich models including CLIP and SWSL blur the boundary, making more human-like errors than standard models.

natural follow-up question is to ask not just how many, but *which* errors models make—i.e., do they make errors on the same individual images as humans on OOD data (an important characteristic of a "human-like" model, cf. [32, 75])? This is quantified via *error consistency* (defined in Section 2); which additionally allows us to compare models with each other, asking e.g. which model classes make similar errors. In Figure 4, we compare all models with each other and with humans, asking whether they make errors on the same images. On this particular dataset (sketch images), we can see one big model cluster. Irrespective of whether one takes a standard supervised model, a self-supervised model, an adversarially trained model or a vision transformer, all those models make highly systematic errors (which extends the results of [32, 76] who found similarities between standard vanilla CNNs). Humans, on the other hand, show a very different pattern of errors. Interestingly, the boundary between humans and some data-rich models at the bottom of the figure—especially CLIP (400M images) and SWSL (940M)—is blurry: some (but not all) data-rich models much more closely mirror the patterns of errors that humans make, and we identified the first models to achieve higher error consistency with humans than with other (standard) models. Are these promising results shared across datasets, beyond the sketch images? In Figures 1c and 1d, aggregated results over 17 datasets are presented. Here, we can see that data-rich models approach human-to-human observed consistency, but not error consistency. Taken in isolation, *observed* consistency is not a good measure of image-level consistency since it does not take consistency by chance into account; *error* consistency tracks whether there is consistency beyond chance; here we see that there is still

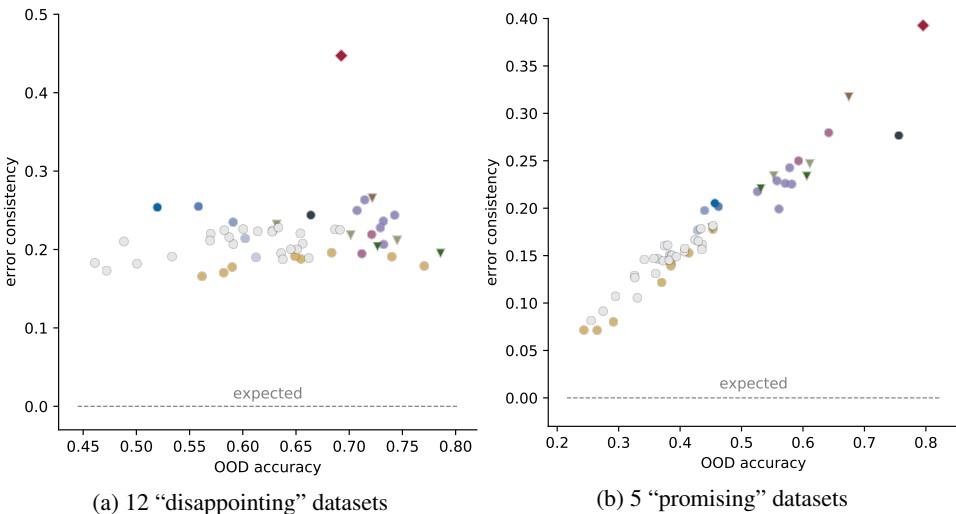

(a) 12 "disappointing" datasets                    (b) 5 "promising" datasets

Figure 5: Partial failure, partial success: Error consistency with humans aggregated over multiple datasets. Left: 12 datasets where model accuracies exceed human accuracies; here, there is still a substantial image-level consistency gap to humans. Right: 5 datasets (sketch, silhouette, edge, cue conflict, low-pass) where humans are more robust. Here, OOD accuracy is a near-perfect predictor of image-level consistency; especially data-rich models (e.g. CLIP, SWSL, BiT) narrow the consistency gap to humans. Symbols indicate architecture type (○ convolutional, ▽ vision transformer, ◇ human).

a substantial image-level *consistency gap* between human and machine vision. However, several models improve over vanilla CNNs, especially BiT-M (trained on 14M images) and CLIP (400M images). This progress is non-trivial; at the same time, there is ample room for future improvement.

How do the findings from Figure 4 (showing nearly human-level error consistency for sketch images) and from Figure 1d (showing a substantial consistency gap when aggregating over 17 datasets) fit together? Upon closer inspection, we discovered that there are two distinct cases. On 12 datasets (stylized, colour/greyscale, contrast, high-pass, phase-scrambling, power-equalisation, false colour, rotation, eidolonI, -II and -III as well as uniform noise), the human-machine gap is large; here, more robust models do not show improved error consistency (as can be see in Figure 5a). On the other hand, for five datasets (sketch, silhouette, edge, cue conflict, low-pass filtering), there is a completely different result pattern: Here, OOD accuracy is a near-perfect predictor of error consistency, which means that improved generalisation robustness leads to more human-like errors (Figure 5b). Furthermore, training on large-scale datasets leads to considerable improvements along both axes for standard CNNs. Within models trained on larger datasets, CLIP scores best; but models with a standard architecture (SWSL: based on ResNet-50 and ResNeXt-101) closely follow suit.

It remains an open question why the training dataset appears to have the most important impact on a model's decision boundary as measured by error consistency (as opposed to other aspects of a model's inductive bias). Datasets contain various shortcut opportunities [14], and if two different models are trained on similar data, they might converge to a similar solution simply by exploiting the same shortcuts—which would also fit well to the finding that adversarial examples typically transfer very well between different models [77, 78]. Making models more flexible (such as transformers, a generalisation of CNNs) wouldn't change much in this regard, since flexible models can still exploit the same shortcuts. Two predictions immediately follow from this hypothesis: (1.) error consistency between two identical models trained on very different datasets, such as ImageNet vs. Stylized-ImageNet, is much lower than error consistency between very different models (ResNet-50 vs. VGG-16) trained on the same dataset. (2.) error consistency between ResNet-50 and a highly flexible model (e.g., a vision transformer) is much higher than error consistency between ResNet-50 and a highly constrained model like BagNet-9 [79]. We provide evidence for both predictions in Appendix B, which makes the shortcut hypothesis of model similarity a potential starting point for future analyses. Looking forward, it may be worth exploring the links between shortcut learning and image difficulty, such as understanding whether many "trivially easy" images in common datasets like ImageNet causes models to expoit the same characteristics irrespective of their architecture [80].

# 5 Discussion

**Summary**   We set out to answer the question: *Are we making progress in closing the gap between human and machine vision?* In order to quantify progress, we performed large-scale psychophysical experiments on 17 out-of-distribution distortion datasets (open-sourced along with evaluation code as a benchmark to track future progress). We then investigated models that push the boundaries of traditional deep learning (different objective functions, architectures, and dataset sizes ranging from 1M to 1B), asking how they perform relative to human visual perception. We found that the OOD distortion robustness gap between human and machine vision is closing, as the best models now match or exceed human accuracies. At the same time, an image-level consistency gap remains; however, this gap that is at least in some cases narrowing for models trained on large-scale datasets.

**Limitations**   Model robustness is studied from many different viewpoints, including adversarial robustness [77], theoretical robustness guarantees [e.g. 81], or label noise robustness [e.g. 82]. The focus of our study is robustness towards non-adversarial out-of-distribution data, which is particularly well-suited for comparisons with humans. Since we aimed at a maximally fair comparison between feedforward models and human perception, presentation times for human observers were limited to 200 ms in order to limit the influence of recurrent processing. Therefore, human ceiling performance might be higher still (given more time); investigating this would mean going beyond "core object recognition", which happens within less than 200 ms during a single fixation [83]. Furthermore, human and machine vision can be compared in many different ways. This includes comparing against neural data [84, 85], contrasting Gestalt effects [e.g. 86], object similarity judgments [87], or mid-level properties [61] and is of course not limited to studying object recognition. By no means do we mean to imply that our behavioural comparison is the only feasible option—on the contrary, we believe it will be all the more exciting to investigate whether our behavioural findings have implications for other means of comparison!

**Discussion**   We have to admit that we view our results concerning the benefits of increasing dataset size by one-to-three orders of magnitude with mixed feelings. On the one hand, "simply" training standard models on (a lot) more data certainly has an intellectually disappointing element—particularly given many rich ideas in the cognitive science and neuroscience literature on which architectural changes might be required to bring machine vision closer to human vision [88–93]. Additionally, large-scale training comes with infrastructure demands that are hard to meet for many academic researchers. On the other hand, we find it truly exciting to see that machine models are closing not just the OOD distortion robustness gap to humans, but that also, at least for some datasets, those models are actually making more human-like decisions on an individual image level; image-level response consistency is a much stricter behavioural requirement than just e.g. matching overall accuracies. Taken together, our results give reason to celebrate partial success in closing the gap between human and machine vision. In those cases where there is still ample room for improvement, our psychophysical benchmark datasets and toolbox may prove useful in quantifying future progress.

## Acknowledgments and disclosure of funding

We thank Andreas Geiger, Simon Kornblith, Kristof Meding, Claudio Michaelis and Ludwig Schmidt for helpful discussions regarding different aspects of this work; Lukas Huber, Maximus Mutschler, David-Elias Künstle for feedback on the manuscript; Ken Kahn for pointing out typos; Santiago Cadena for sharing a PyTorch implementation of SimCLR; Katherine Hermann and her collaborators for providing supervised SimCLR baselines; Uli Wannek and Silke Gramer for infrastructure/administrative support; the many authors who made their models publicly available; and our anonymous reviewers for many valuable suggestions.

Furthermore, we are grateful to the International Max Planck Research School for Intelligent Systems (IMPRS-IS) for supporting R.G.; the Collaborative Research Center (Projektnummer 276693517—SFB 1233: Robust Vision) for supporting M.B. and F.A.W. This work was supported by the German Federal Ministry of Education and Research (BMBF): Tübingen AI Center, FKZ: 01IS18039A (W.B. and M.B.). F.A.W. is a member of the Machine Learning Cluster of Excellence, EXC number 2064/1—Project number 390727645. M.B. and W.B. acknowledge funding from the MICrONS program of the Intelligence Advanced Research Projects Activity (IARPA) via Department of Interior/Interior Business Center (DoI/IBC) contract number D16PC00003. W.B. acknowledges financial support via the Emmy Noether Research Group on The Role of Strong Response Consistency for Robust and Explainable Machine Vision funded by the German Research Foundation (DFG) under grant no. BR 6382/1-1.

## Author contributions

Project idea: R.G. and W.B.; project lead: R.G.; coding toolbox and model evaluation pipeline: R.G., K.N. and B.M. based on a prototype by R.G.; training models: K.N. with input from R.G., W.B. and M.B.; data visualisation: R.G., B.M. and K.N. with input from M.B., F.A.W. and W.B.; psychophysical data collection: T.T. (12 datasets) and B.M. (2 datasets) under the guidance of R.G. and F.A.W.; curating stimuli: R.G.; interpreting analyses and findings: R.G., M.B., F.A.W. and W.B.; guidance, feedback, infrastructure & funding acquisition: M.B., F.A.W. and W.B.; paper writing: R.G. with help from F.A.W. and W.B. and input from all other authors.

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
