# Appendix

We here provide details on models (A), describe additional predictions and experiments regarding error consistency mentioned in Section 4 (B), report experimental details regarding our psychphysical experiments (C), provide license information (D), and describe training with ImageNet labels provided by CLIP (E) as well as experiments with supervised SimCLR baseline models (F), provide overall benchmark scores ranking different models (G), describe a regression model (H) and motivate the choice of behavioural response mapping (I). Stimuli are visualized in Figures 7 and 8.

Our Python library,"modelvshuman", to test and benchmark models against high-quality human psychophyiscal data is available from `https://github.com/bethgelab/model-vs-human/`.

## A  Model details

**Standard supervised models.**  We used all 24 available pre-trained models from the PyTorch model zoo version 1.4.0 (VGG: with batch norm).

**Self-supervised models.**  InsDis [38], MoCo [39], MoCoV2 [40], PIRL [41] and InfoMin [42] were obtained as pre-trained models from the PyContrast model zoo. We trained one linear classifier per model on top of the self-supervised representation. A PyTorch [94] implementation of SimCLR [43] was obtained via simclr-converter. All self-supervised models use a ResNet-50 architecture and a different training approach within the framework of contrastive learning [e.g. 95].

**Adversarially trained models.**  We obtained five adversarially trained models [46] from the robust-models-transfer repository. All of them have a ResNet-50 architecture, but a different accuracy-L2-robustness tradeoff indicated by $\epsilon$. Here are the five models that we used, in increasing order of adversarial robustness: $\epsilon = 0, 0.5, 1.0, 3.0, 5.0$.

**Vision transformers.**  Three ImageNet-trained vision transformer (ViT) models [47] were obtained from pytorch-image-models [48]. Specifically, we used vit_small_patch16_224, vit_base_patch16_224 and vit_large_patch16_224. They are referred to as ViT-S, ViT-B and ViT-L throughout the paper. Additionally, we included two transformers that were pre-trained on ImageNet21K [96], i.e. 14M images with some 21K classes, before they were fine-tuned on "standard" ImageNet-1K. These two models are referred to as ViT-L (14M) and ViT-B (14M) in the paper. They were obtained from the PyTorch-Pretrained-ViT repository, where they are called L_16_imagenet1k and B_16_imagenet1k. (No ViT-S model was available from the repository.) Note that the "imagenet1k" suffix in the model names does not mean the model was only trained on ImageNet1K. On the contrary, this indicates fine-tuning on ImageNet; as mentioned above these models were pre-trained on ImageNet21K before fine-tuning.

**CLIP.**  OpenAI trained a variety of CLIP models using different backbone networks [51]. Unfortunately, the best-performing model has not been released so far, and it is not currently clear whether it will be released at some point according to issue #2 of OpenAI's CLIP github repository. We included the most powerful released model in our analysis, a model with a ViT-B/32 backbone.

**Noisy Student**  One pre-trained Noisy Student model was obtained from pytorch-image-models [48], where the model is called tf_efficientnet_e2_ns_475. This involved the following preprocessing (taken from [97]):

```python
from PIL.Image import Image
from torchvision.transforms import Compose, Resize, CenterCrop, ToTensor, Normalize

def get_noisy_student_preprocessing():
    normalize = Normalize(mean=[0.485, 0.456, 0.406],
                          std=[0.229, 0.224, 0.225])
    img_size = 475
    crop_pct = 0.936
    scale_size = img_size / crop_pct
    return Compose([
        Resize(scale_size, interpolation=PIL.Image.BICUBIC),
        CenterCrop(img_size),
        ToTensor(),
        normalize,
    ])
```

**SWSL**  Two pre-trained SWSL (semi-weakly supervised) models were obtained from semi-supervised-ImageNet1K-models, one with a ResNet-50 architecture and one with a ResNeXt101_32x16d architecture.

**BiT-M**  Six pre-trained Big Transfer models were obtained from pytorch-image-models [48], where they are called resnetv2_50x1_bitm, resnetv2_50x3_bitm, resnetv2_101x1_bitm, resnetv2_101x3_bitm, resnetv2_152x2_bitm and resnetv2_152x4_bitm.

**Linear classifier training procedure.**  The PyContrast repository by Yonglong Tian contains a Pytorch implementation of unsupervised representation learning methods, including pre-trained representation weights. The repository provides training and evaluation pipelines, but it supports only multi-node distributed training and does not (currently) provide weights for the classifier. We have used the repository's linear classifier evaluation pipeline to train classifiers for InsDis [38], MoCo [39], MoCoV2 [40], PIRL [41] and InfoMin [42] on ImageNet. Pre-trained weights of the model representations (without classifier) were taken from the provided Dropbox link and we then ran the training pipeline on a NVIDIA TESLA P100 using the default parameters configured in the pipeline. Detailed documentation about running the pipeline and parameters can be found in the PyContrast repository (commit #3541b82).

# B  Error consistency predictions

Table 1: Error consistency across all five non-parametric datasets. Specifically, this comparison compares the influence of dataset vs. architecture (top) and the influence of flexibility vs. constraints (bottom). Results are described in Section B.

|  | sketch | stylized | edge | silhouette | cue conflict |
|---|---|---|---|---|---|
| ResNet-50 vs. VGG-16 | **0.74** | **0.56** | **0.68** | **0.71** | **0.59** |
| ResNet-50 vs. ResNet-50 trained on Stylized-ImageNet | 0.44 | 0.09 | 0.10 | 0.67 | 0.27 |
| ResNet-50 vs. vision transformer (ViT-S) | **0.67** | **0.43** | **0.41** | **0.68** | **0.48** |
| ResNet-50 vs. BagNet-9 | 0.31 | 0.17 | 0.32 | 0.14 | 0.44 |

In Section 4, we hypothesised that shortcut opportunities in the dataset may be a potential underlying cause of high error consistency between models, since all sufficiently flexible models will pick up on those same shortcuts. We then made two predictions which we test here.

**Dataset vs. architecture.**  *Prediction:* error consistency between two identical models trained on very different datasets, such as ImageNet vs. Stylized-ImageNet, is much lower than error consistency between very different models (ResNet-50 vs. VGG-16) trained on the same dataset. *Observation:* According to Table 1, this is indeed the case—training ResNet-50 on a different dataset, Stylized-ImageNet [17], leads to lower error consistency than comparing two ImageNet-trained CNNs with different architecture. While this relationship is not perfect (e.g., the difference is small for silhouette images), we have confirmed that this is a general pattern not limited to the specific networks in the table.

**Flexibility vs. constraints.**  *Prediction:* error consistency between ResNet-50 and a highly flexible model (e.g., a vision transformer) is much higher than error consistency between ResNet-50 and a highly constrained model like BagNet-9 [79]. *Observation:* A vision transformer (ViT-S) indeed shows higher error consistency with ResNet-50 than with BagNet-9 (see Table 1). However, this difference is not large for one out of five datasets (cue conflict). One could imagine different reasons for this: perhaps BagNet-9 is still flexible enough to learn a decision rule close to the one of standard ResNet-50 for cue conflict images; and of course there is also the possibility that the hypothesis is wrong. Further insights could be gained by testing successively more constrained versions of the same base model.

# C   Experimental details regarding psychophysical experiments

## C.1   Participant instructions and preparation

Participants were explained how to respond (via mouse click), instructed to respond as accurately as possible, and to go with their best guess if unsure. In order to rule out any potential misunderstanding, participants were asked to name all 16 categories on the response screen. Prior to the experiment, visual acuity was measured with a Snellen chart to ensure normal or corrected to normal vision. Furthermore, four blocks of 80 practice trials each (320 practice trials in total) on undistorted colour or greyscale images were conducted (non-overlapping with experimental stimuli) to gain familiarity with the task. During practice trials, but not experimental trials, visual and auditory feedback was provided: the correct category was highlighted and a "beep" sound was played for incorrect or missed trials. The experiment itself consisted of blocks of 80 trials each, after each blocks participants were free to take a break. In order to increase participant motivation, aggregated performance over the last block was displayed on the screen.

## C.2   Participant risks

Our experiment was a standard perceptual experiment, for which no IRB approval was required. The task consisted of viewing corrupted images and clicking with a computer mouse. In order to limit participant risks related to a COVID-19 infection, we implemented the following measures: (1.) The experimenter was tested for corona twice per week. (2.) Prior to participation in our experiments, participants were explained that they could perform a (cost-free) corona test next to our building, and that if they choose to do so, we would pay them 10€/hour for the time spent doing the test and waiting for the result (usually approx. 15–30min). (3.) Experimenter and participant adhered to a strict distance of at least 1.5m during the entire course of the experiment, including instructions and practice trials. During the experiment itself, the participant was the only person in the room; the experimenter was seated in an adjacent room. (4.) Wearing a medical mask was mandatory for both experimenter and participant. (5.) Participants were asked to disinfect their hands prior to the experiment; additionally the desk, mouse etc. were disinfected after completion of an experiment. (6.) Participants were tested in a room where high-performance ventilation was installed; in order to ensure that the ventilation was working as expected we performed a one-time safety check measuring $CO_2$ parts-per-million before we decided to go ahead with the experiments.

## C.3   Participant remuneration

Participants were paid 10€ per hour or granted course credit. Additionally, an incentive bonus of up to 15€ could be earned on top of the standard remuneration. This was meant to further motivate our participants to achieve their optimal performance. The minimum performance for receiving a bonus was set as 15% below the mean of the previous experiments accuracy. The bonus then was linearly calculated with the maximal bonus being given from 15% above the previous experiments mean. The total amount spent on participant compensation amounts to 647,50€.

## C.4   Participant declaration of consent

Participants were asked to review and sign the following declaration of consent (of which they received a copy):

*Psychophysical study*
*Your task consists of viewing visual stimuli on a computer monitor and evaluating them by pressing a key. Participation in a complete experimental session is remunerated at 10 Euros/hour.*
*Declaration of consent*
*Herewith I agree to participate in a behavioural experiment to study visual perception. My participation in the study is voluntary. I am informed that I can stop the experiment at any time and without giving any reason without incurring any disadvantages. I know that I can contact the experimenter at any time with questions about the research project.*
*Declaration of consent for data processing and data publication*
*Herewith I agree that the experimental data obtained in the course of the experiment may be used in semianonymised form for scientific evaluation and publication. I agree that my personal data (e.g.*

*name, phone number, address) may be stored in digital form; they will not be used for any other purpose than for contacting me. This personal data will remain exclusively within the Wichmannlab and will not be passed on to third parties at any time.*

## D   Licenses

Licenses for datasets, code and models are included in our code (see directory "licenses/", file "LICENSES_OVERVIEW.md" of `https://github.com/bethgelab/model-vs-human`.

## E   Training with CLIP labels

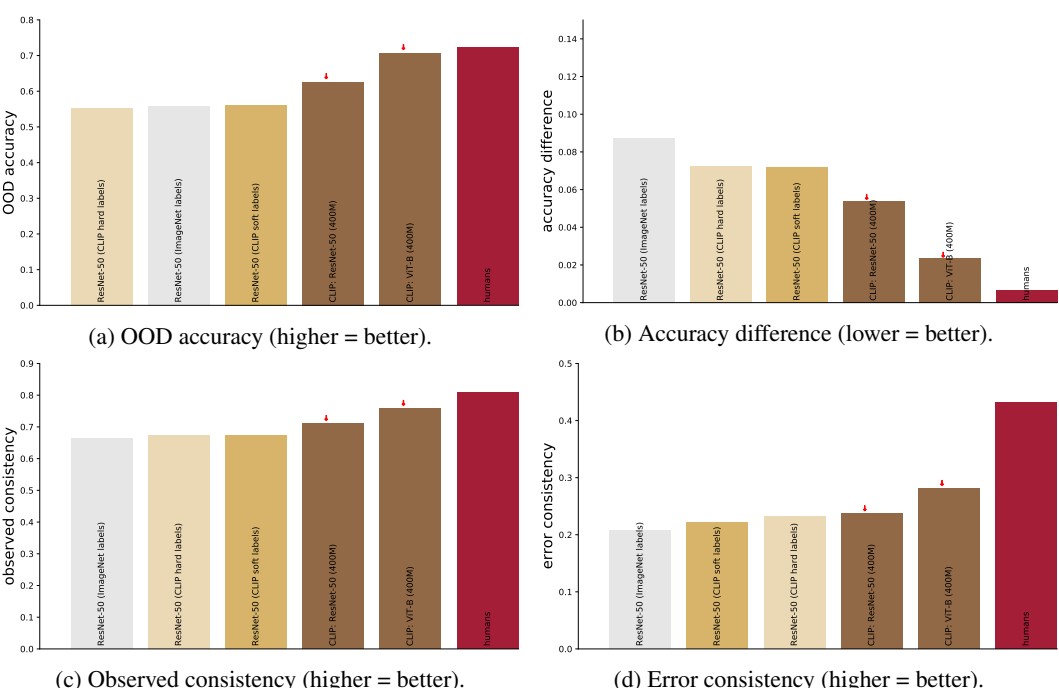

(a) OOD accuracy (higher = better).

(b) Accuracy difference (lower = better).

(c) Observed consistency (higher = better).

(d) Error consistency (higher = better).

Figure 6: Aggregated results comparing models with and without CLIP-provided labels. Comparison of standard ResNet-50 (light grey), CLIP with vision transformer backend (brown), CLIP with ResNet-50 backend (brown), and standard ResNet-50 with hard labels (bright yellow) vs. soft labels (dark yellow) provided by evaluating standard CLIP on ImageNet; as well as humans (red diamonds) for comparison. Detailed performance across datasets in Figure 16.

As CLIP performed very well across metrics, we intended to obtain a better understanding for why this might be the case. One hypothesis is that CLIP might just receive better labels: About 6% of ImageNet validation images are mis-labeled according to Northcutt et al. [74]. We therefore designed an experiment where we re-labeled the entire ImageNet training and validation dataset using CLIP predictions as ground truth (`https://github.com/kantharajucn/CLIP-imagenet-evaluation`). Having re-labeled ImageNet, we then trained a standard ResNet-50 model from scratch on this dataset using the standard PyTorch ImageNet training script. Training was performed on our on-premise cloud using four RTX 2080 Ti GPUs for five days. We ran the training pipeline in distributed mode with an nccl backed using the default parameters configured in the script, except for the number of workers which we changed to 25. Cross-entropy loss was used to train two models, once with CLIP hard labels (the top-1 class predicted by CLIP) and once with CLIP soft labels (using CLIP's full posterior distribution as training target). The accuracies on the original ImageNet validation dataset of the resulting models ResNet50-CLIP-hard-labels and ResNet-50-CLIP-soft-labels are 63.53 (top-1), 86.97 (top-5) and 64.63 (top-1), 88.60 (top-5) respectively. In order to make sure that the model trained on soft labels had indeed learned to approximate CLIP's posterior distribution on ImageNet, we calculated the KL divergence between CLIP soft labels and probability distributions

from ResNet-50 trained on the CLIP soft labels. The resulting value of 0.001 on both ImageNet training and validation dataset is sufficiently small to conclude that the model had successfully learned to approximate CLIP's posterior distribution on ImageNet. The results are visualised in Figure 6. The results indicate that simply training a standard ResNet-50 model with labels provided by CLIP does not lead to strong improvements on any metric, which means that ImageNet label errors are unlikely to hold standard models back in terms of OOD accuracy and consistency with human responses.

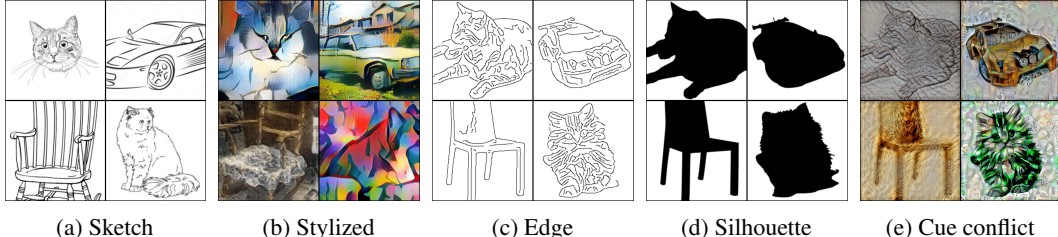

| (a) Sketch | (b) Stylized | (c) Edge | (d) Silhouette | (e) Cue conflict |

Figure 7: Exemplary stimuli (nonparametric image manipulations) for the following datasets: sketch (7 observers, 800 trials each), stylized (5 observers, 800 trials each), edge (10 observers, 160 trials each), silhouette (10 observers, 160 trials each), and cue conflict (10 observers, 1280 trials each). Figures c–e reprinted from [32] with permission from the authors. [32] also analyzed "diagnostic" images, i.e. stimuli that most humans correctly classified (but few networks) and vice-versa.

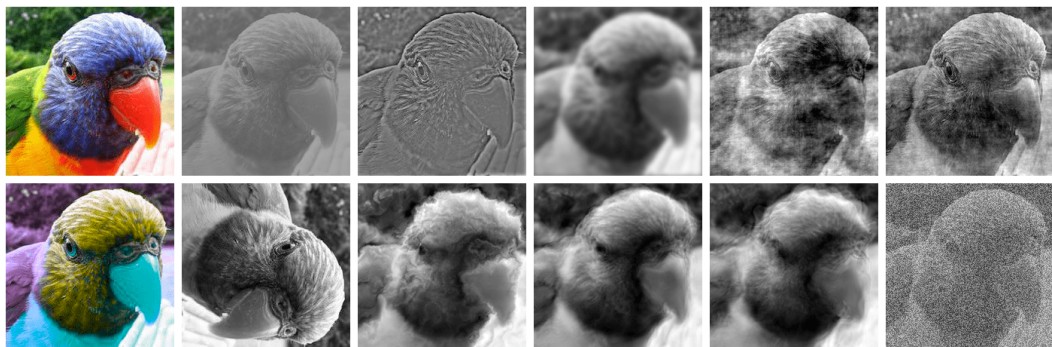

Figure 8: Exemplary stimuli (parametric image manipulations). Manipulations are either binary (e.g. colour vs. grayscale) or they have a parameter (such as the degree of rotation, or the contrast level). Top row: colour vs. grayscale (4 observers, 1280 trials each), low contrast (4 observers, 1280 trials each), high-pass (4 observers, 1280 trials each), low-pass/blurring (4 observers, 1280 trials each), phase noise (4 observers, 1120 trials each), true power spectrum vs. power equalisation (4 observers, 1120 trials each). Bottom row: true vs. opponent colour (4 observers, 1120 trials each), rotation (4 observers, 1280 trials each), Eidolon I (4 observers, 1280 trials each), Eidolon II (4 observers, 1280 trials each), Eidolon III (4 observers, 1280 trials each, additive uniform noise (4 observers, 1280 trials each). Figure adapted from [33] with permission from the authors.

## F   Supervised SimCLR baseline models

Figure 15 compares the noise generalisation performance self-supervised SimCLR models against augmentation-matched baseline models. The results indicate that the superior performance of SimCLR in Figure 2 are largely a consequence of SimCLR's data augmentation scheme, rather than a property of the self-supervised contrastive loss.

## G   Benchmark scores

Figure 1 in the main paper shows aggregated scores for the most robust model in terms of OOD accuracy (Figure 1a), and for the most human-like models in terms of accuracy, observed and error consistency (Figures 1b, 1c, 1d). Numerically, these metrics are represented in two tables,

ranking the models according to out-of-distribution robustness (Table 3) and human-like behaviour (Table 2). Since the latter is represented by three different metrics (each characterising a distinct aspect), the mean rank across those three metrics is used to obtain a final ordering. The following conditions and datasets influence benchmark scores: For the five nonparametric datasets, all datasets are taken into account. For the twelve parametric datasets, we also take all datasets into account (overall, all 17 datasets are weighted equally); however, we exclude certain conditions for principled reasons. First of all, the easiest condition is always excluded since it does not test out-of-distribution behaviour (e.g., for the contrast experiment, 100% contrast is more of a baseline condition rather than a condition of interest). Furthermore, we exclude all conditions for which human average performance is strictly smaller than 0.2, since e.g. comparisons against human error patterns are futile if humans are randomly guessing since they cannot identify the stimuli anymore. For these reasons, the following conditions are not taken into account when computing the benchmark scores. Colour vs. greyscale experiment: condition "colour". True vs. false colour experiment: condition "true colour". Uniform noise experiment: conditions 0.0, 0.6, 0.9. Low-pass experiment: conditions 0, 15, 40. Contrast experiment: conditions 100, 3, 1. High-pass experiment: conditions inf, 0.55, 0.45, 0.4. Eidolon I experiment: conditions 0, 6, 7. Phase noise experiment: conditions 0, 150, 180. Eidolon II experiment: conditions 0, 5, 6, 7. Power-equalisation experiment: condition "original power spectrum". Eidolon III experiment: conditions 0, 4, 5, 6, 7. Rotation experiment: condition 0.

# H   Regression model

In order to quantify the influence of known independent variables (architecture: transformers vs. ConvNets; data: small (ImageNet) vs. large ("more" than standard ImageNet); objective: supervised vs. self-supervised) on known dependent variables (OOD accuracy and error consistency with humans), we performed a regression analysis using R version 3.6.3 (functions `lm` for fitting and `anova` for regression model comparison). We modelled the influence of those predictors on OOD accuracy, and on error consistency with human observers in two separate linear regression models (one per dependent variable). To this end, we used incremental model building, i.e. starting with one significant predictor and subsequently adding predictors if the reduction of degrees of freedom is justified by a significantly higher degree of explained variance (alpha level: .05). Both error consistency and accuracy, for our 52 models, followed (approximately) a normal distribution, as confirmed by density and Q-Q-plots. That being said, the fit was better for error consistency than for accuracy.

The final regression model for error consistency showed:

- a significant main effect for transformers over CNNs (p = 0.01936 *),
- a significant main effect for large datasets over small datasets (p = 3.39e-05 ***),
- a significant interaction between dataset size and objective function (p = 0.00625 **),
- no significant main effect of objective function (p = 0.10062, n.s.)

Residual standard error: 0.02156 on 47 degrees of freedom
Multiple R-squared: 0.5045, Adjusted R-squared: 0.4623
F-statistic: 11.96 on 4 and 47 DF, p-value: 8.765e-07
Significance codes: 0 '***' 0.001 '**' 0.01 '*' 0.05

The final regression model for OOD accuracy showed:

- a significant main effect for large datasets over small datasets (p = 4.65e-09 ***),
- a significant interaction between dataset size and architecture type (transformer vs. CNNs; p = 0.0174 *),
- no significant main effect for transformers vs. CNNs (p = 0.8553, n.s.)

Residual standard error: 0.0593 on 48 degrees of freedom
Multiple R-squared: 0.5848, Adjusted R-squared: 0.5588
F-statistic: 22.53 on 3 and 48 DF, p-value: 3.007e-09
Significance codes: 0 '***' 0.001 '**' 0.01 '*' 0.05

Limitations: a linear regression model can only capture linear effects; furthermore, diagnostic plots showed a better fit for the error consistency model (where residuals roughly followed the expected distribution as confirmed by a Q-Q-plot) than for the OOD accuracy model (where residuals were not perfectly normal distributed).

## I  Mapping behavioural decisions

Comparing model and human classification decisions comes with a challenge: we simply cannot ask human observers to classify objects into 1,000 classes (as for standard ImageNet models). Even if this were feasible in terms of experimental time constraints, most humans don't routinely know the names of a hundred different dog breeds. What they do know, however, is how to tell dogs apart from cats and from airplanes, chairs and boats. Those are so-called "basic" or "entry-level" categories [98]. In line with previous work [17, 32, 33], we therefore used a set of 16 basic categories in our experiments. For ImageNet-trained models, to obtain a choice from the same 16 categories, the 1,000 class decision vector was mapped to those 16 classes using the WordNet hierarchy [34]. Those 16 categories were chosen to reflect a large chunk of ImageNet (227 classes, i.e. roughly a quarter of all ImageNet categories is represented by those 16 basic categories). In order to obtain classification decisions from ImageNet-trained models for those 16 categories, at least two choices are conceivable: re-training the final classification layer or using a principled mapping. Since any training involves making a number of choices (hyperparameters, optimizer, dataset, ...) that may potentially influence and in the worst case even bias the results (e.g. for ShuffleNet, more than half of the model's parameters are contained in the final classification layer!), we decided against training and for a principled mapping by calculating the probability of a coarse class as the average of the probabilities of the corresponding fine-grained classes. Why is this mapping principled? As derived by [33] (pages 22 and 23 in the Appendix of the arXiv version, `https://arxiv.org/pdf/1808.08750.pdf`), this is the optimal way to map (i.e. aggregate) probabilities from many fine-grained classes to a few coarse classes. Essentially, the aggregation can be derived by calculating the posterior distribution of a discriminatively trained CNN under a new prior chosen at test time (here: 1/16 over coarse classes).

Table 2: Benchmark table of model results. The three metrics "accuracy difference" "observed consistency" and "error consistency" (plotted in Figure 1) each produce a different model ranking. The mean rank of a model across those three metrics is used to rank the models on our benchmark.

| model | accuracy diff. ↓ | obs. consistency ↑ | error consistency ↑ | mean rank ↓ |
|---|---|---|---|---|
| CLIP: ViT-B (400M) | **0.023** | 0.758 | **0.281** | **1.333** |
| SWSL: ResNeXt-101 (940M) | 0.028 | 0.752 | 0.237 | 4.000 |
| BiT-M: ResNet-101x1 (14M) | 0.034 | 0.733 | 0.252 | 4.333 |
| BiT-M: ResNet-152x2 (14M) | 0.035 | 0.737 | 0.243 | 5.000 |
| ViT-L | 0.033 | 0.738 | 0.222 | 6.667 |
| BiT-M: ResNet-152x4 (14M) | 0.035 | 0.732 | 0.233 | 7.667 |
| BiT-M: ResNet-50x3 (14M) | 0.040 | 0.726 | 0.228 | 9.333 |
| BiT-M: ResNet-50x1 (14M) | 0.042 | 0.718 | 0.240 | 9.667 |
| ViT-L (14M) | 0.035 | 0.744 | 0.206 | 9.667 |
| SWSL: ResNet-50 (940M) | 0.041 | 0.727 | 0.211 | 11.667 |
| ViT-B | 0.044 | 0.719 | 0.223 | 12.000 |
| BiT-M: ResNet-101x3 (14M) | 0.040 | 0.720 | 0.204 | 14.333 |
| densenet201 | 0.060 | 0.695 | 0.212 | 15.000 |
| ViT-B (14M) | 0.049 | 0.717 | 0.209 | 15.000 |
| ViT-S | 0.066 | 0.684 | 0.216 | 16.667 |
| densenet169 | 0.065 | 0.688 | 0.207 | 17.333 |
| inception_v3 | 0.066 | 0.677 | 0.211 | 17.667 |
| Noisy Student: ENetL2 (300M) | 0.040 | **0.764** | 0.169 | 18.000 |
| ResNet-50 L2 eps 1.0 | 0.079 | 0.669 | 0.224 | 21.000 |
| ResNet-50 L2 eps 3.0 | 0.079 | 0.663 | 0.239 | 22.000 |
| wide_resnet101_2 | 0.068 | 0.676 | 0.187 | 24.333 |
| SimCLR: ResNet-50x4 | 0.071 | 0.698 | 0.179 | 24.667 |
| SimCLR: ResNet-50x2 | 0.073 | 0.686 | 0.180 | 25.333 |
| ResNet-50 L2 eps 0.5 | 0.078 | 0.668 | 0.203 | 25.333 |
| densenet121 | 0.077 | 0.671 | 0.200 | 25.333 |
| resnet101 | 0.074 | 0.671 | 0.192 | 25.667 |
| resnet152 | 0.077 | 0.675 | 0.190 | 25.667 |
| resnext101_32x8d | 0.074 | 0.674 | 0.182 | 26.667 |
| ResNet-50 L2 eps 5.0 | 0.087 | 0.649 | 0.240 | 27.000 |
| resnet50 | 0.087 | 0.665 | 0.208 | 28.667 |
| resnet34 | 0.084 | 0.662 | 0.205 | 29.333 |
| vgg19_bn | 0.081 | 0.660 | 0.200 | 30.000 |
| resnext50_32x4d | 0.079 | 0.666 | 0.184 | 30.333 |
| SimCLR: ResNet-50x1 | 0.080 | 0.667 | 0.179 | 32.000 |
| resnet18 | 0.091 | 0.648 | 0.201 | 34.667 |
| vgg16_bn | 0.088 | 0.651 | 0.198 | 34.667 |
| wide_resnet50_2 | 0.084 | 0.663 | 0.176 | 35.667 |
| MoCoV2: ResNet-50 | 0.083 | 0.660 | 0.177 | 36.000 |
| mobilenet_v2 | 0.092 | 0.645 | 0.196 | 37.000 |
| ResNet-50 L2 eps 0.0 | 0.086 | 0.654 | 0.178 | 37.333 |
| mnasnet1_0 | 0.092 | 0.646 | 0.189 | 38.333 |
| vgg11_bn | 0.106 | 0.635 | 0.193 | 38.667 |
| InfoMin: ResNet-50 | 0.086 | 0.659 | 0.168 | 39.333 |
| vgg13_bn | 0.101 | 0.631 | 0.180 | 41.000 |
| mnasnet0_5 | 0.110 | 0.617 | 0.173 | 45.000 |
| MoCo: ResNet-50 | 0.107 | 0.617 | 0.149 | 47.000 |
| alexnet | 0.118 | 0.597 | 0.165 | 47.333 |
| squeezenet1_1 | 0.131 | 0.593 | 0.175 | 47.667 |
| PIRL: ResNet-50 | 0.119 | 0.607 | 0.141 | 48.667 |
| shufflenet_v2_x0_5 | 0.126 | 0.592 | 0.160 | 49.333 |
| InsDis: ResNet-50 | 0.131 | 0.593 | 0.138 | 50.667 |
| squeezenet1_0 | 0.145 | 0.574 | 0.153 | 51.000 |

Table 3: Benchmark table of model results (accuracy).

| model | OOD accuracy ↑ | rank ↓ |
|---|---|---|
| Noisy Student: ENetL2 (300M) | **0.829** | **1.000** |
| ViT-L (14M) | 0.733 | 2.000 |
| CLIP: ViT-B (400M) | 0.708 | 3.000 |
| ViT-L | 0.706 | 4.000 |
| SWSL: ResNeXt-101 (940M) | 0.698 | 5.000 |
| BiT-M: ResNet-152x2 (14M) | 0.694 | 6.000 |
| BiT-M: ResNet-152x4 (14M) | 0.688 | 7.000 |
| BiT-M: ResNet-101x3 (14M) | 0.682 | 8.000 |
| BiT-M: ResNet-50x3 (14M) | 0.679 | 9.000 |
| SimCLR: ResNet-50x4 | 0.677 | 10.000 |
| SWSL: ResNet-50 (940M) | 0.677 | 11.000 |
| BiT-M: ResNet-101x1 (14M) | 0.672 | 12.000 |
| ViT-B (14M) | 0.669 | 13.000 |
| ViT-B | 0.658 | 14.000 |
| BiT-M: ResNet-50x1 (14M) | 0.654 | 15.000 |
| SimCLR: ResNet-50x2 | 0.644 | 16.000 |
| densenet201 | 0.621 | 17.000 |
| densenet169 | 0.613 | 18.000 |
| SimCLR: ResNet-50x1 | 0.596 | 19.000 |
| resnext101_32x8d | 0.594 | 20.000 |
| resnet152 | 0.584 | 21.000 |
| wide_resnet101_2 | 0.583 | 22.000 |
| resnet101 | 0.583 | 23.000 |
| ViT-S | 0.579 | 24.000 |
| densenet121 | 0.576 | 25.000 |
| MoCoV2: ResNet-50 | 0.571 | 26.000 |
| inception_v3 | 0.571 | 27.000 |
| InfoMin: ResNet-50 | 0.571 | 28.000 |
| resnext50_32x4d | 0.569 | 29.000 |
| wide_resnet50_2 | 0.566 | 30.000 |
| resnet50 | 0.559 | 31.000 |
| resnet34 | 0.553 | 32.000 |
| ResNet-50 L2 eps 0.5 | 0.551 | 33.000 |
| ResNet-50 L2 eps 1.0 | 0.547 | 34.000 |
| vgg19_bn | 0.546 | 35.000 |
| ResNet-50 L2 eps 0.0 | 0.545 | 36.000 |
| ResNet-50 L2 eps 3.0 | 0.530 | 37.000 |
| vgg16_bn | 0.530 | 38.000 |
| mnasnet1_0 | 0.524 | 39.000 |
| resnet18 | 0.521 | 40.000 |
| mobilenet_v2 | 0.520 | 41.000 |
| MoCo: ResNet-50 | 0.502 | 42.000 |
| ResNet-50 L2 eps 5.0 | 0.501 | 43.000 |
| vgg13_bn | 0.499 | 44.000 |
| vgg11_bn | 0.498 | 45.000 |
| PIRL: ResNet-50 | 0.489 | 46.000 |
| mnasnet0_5 | 0.472 | 47.000 |
| InsDis: ResNet-50 | 0.468 | 48.000 |
| shufflenet_v2_x0_5 | 0.440 | 49.000 |
| alexnet | 0.434 | 50.000 |
| squeezenet1_1 | 0.425 | 51.000 |
| squeezenet1_0 | 0.401 | 52.000 |

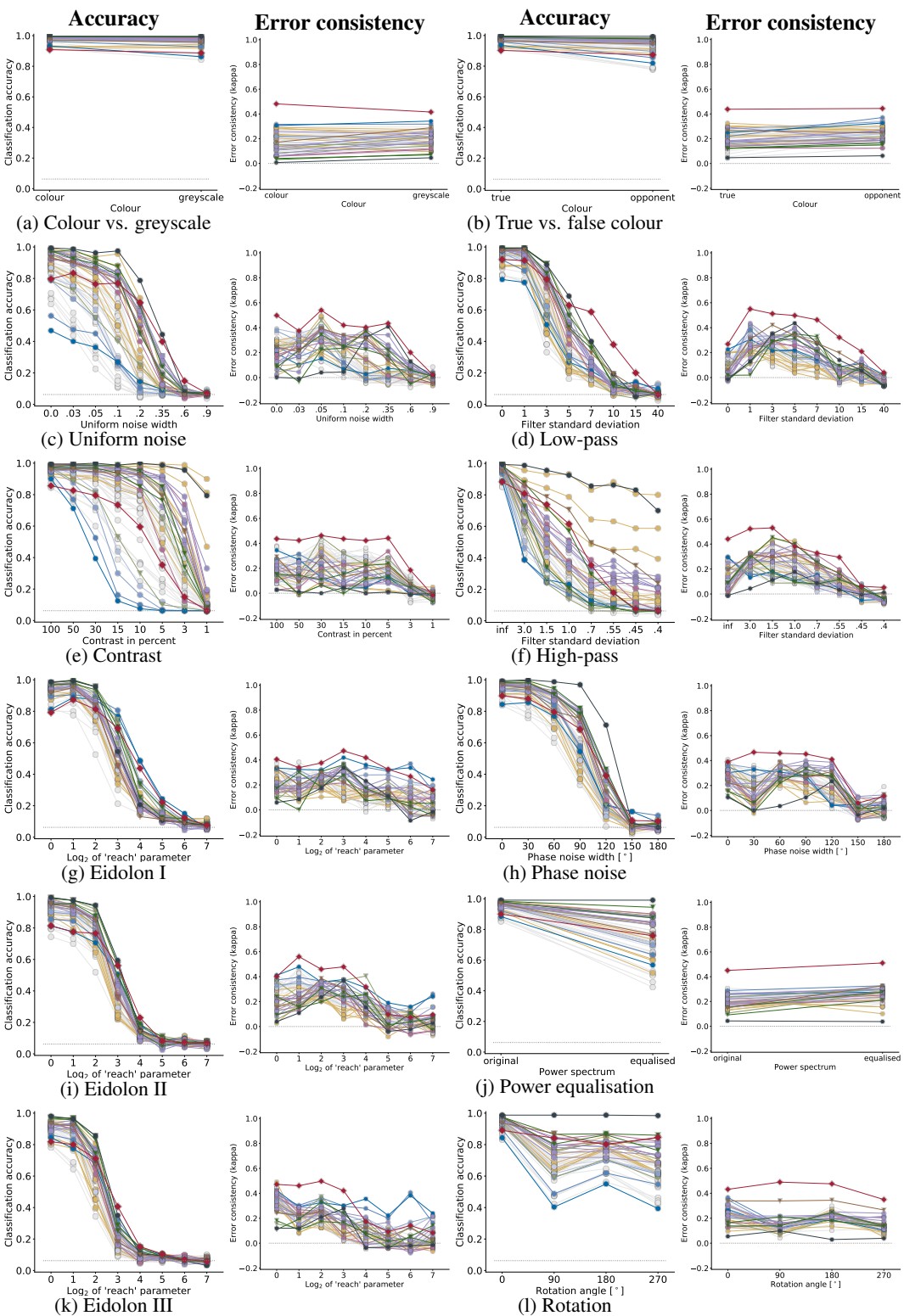

Figure 9: OOD generalisation and error consistency results for humans, standard supervised CNNs, self-supervised models, adversarially trained models, vision transformers, noisy student, BiT, SWSL, CLIP. Symbols indicate architecture type (○ convolutional, ▽ vision transformer, ◇ human); best viewed on screen. 'Accuracy' measures recognition performance (higher is better), 'error consistency' how closely image-level errors are aligned with humans. Accuracy results are identical to Figure 2 in the main paper. In many cases, human-to-human error consistency increases for moderate distortion levels and drops afterwards.

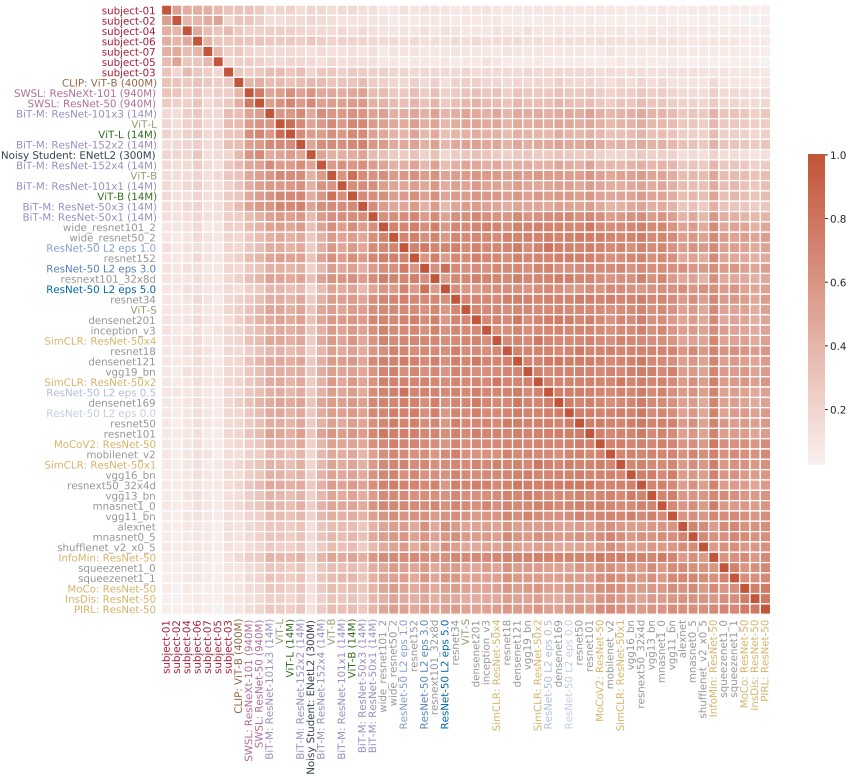

Figure 10: Error consistency for 'sketch' images (same as Figure 4 but sorted w.r.t. mean error consistency with humans).

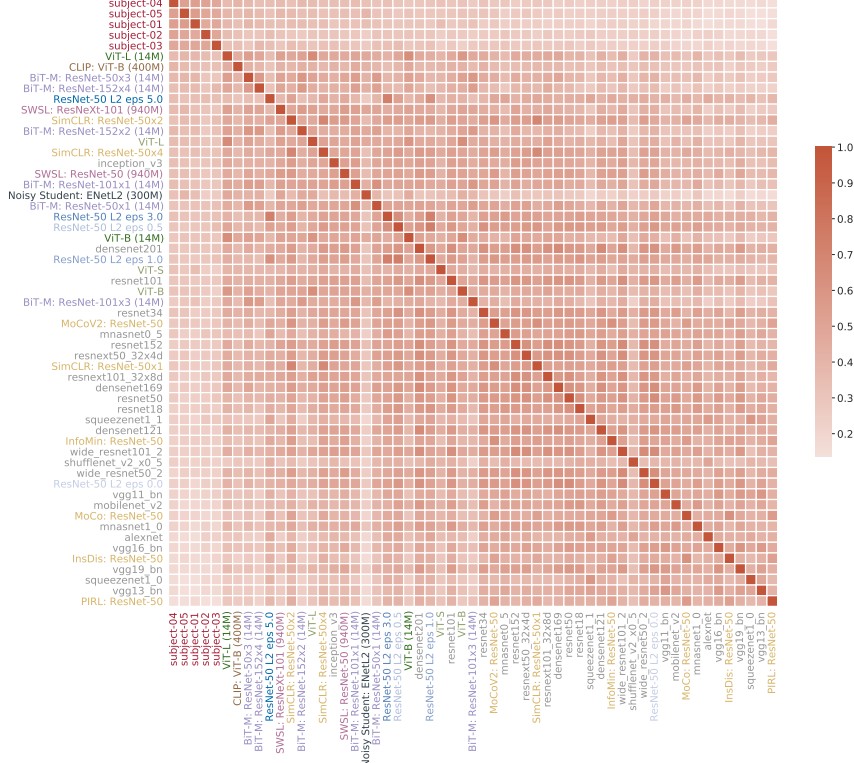

Figure 11: Error consistency for 'stylized' images (sorted w.r.t. mean error consistency with humans).

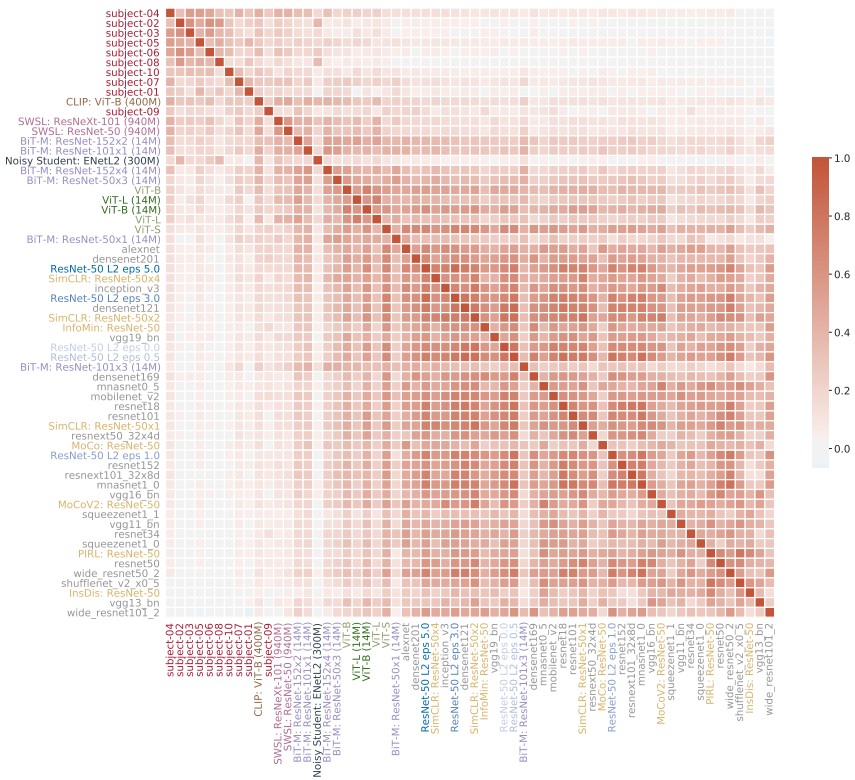

Figure 12: Error consistency for 'edge' images (sorted w.r.t. mean error consistency with humans).

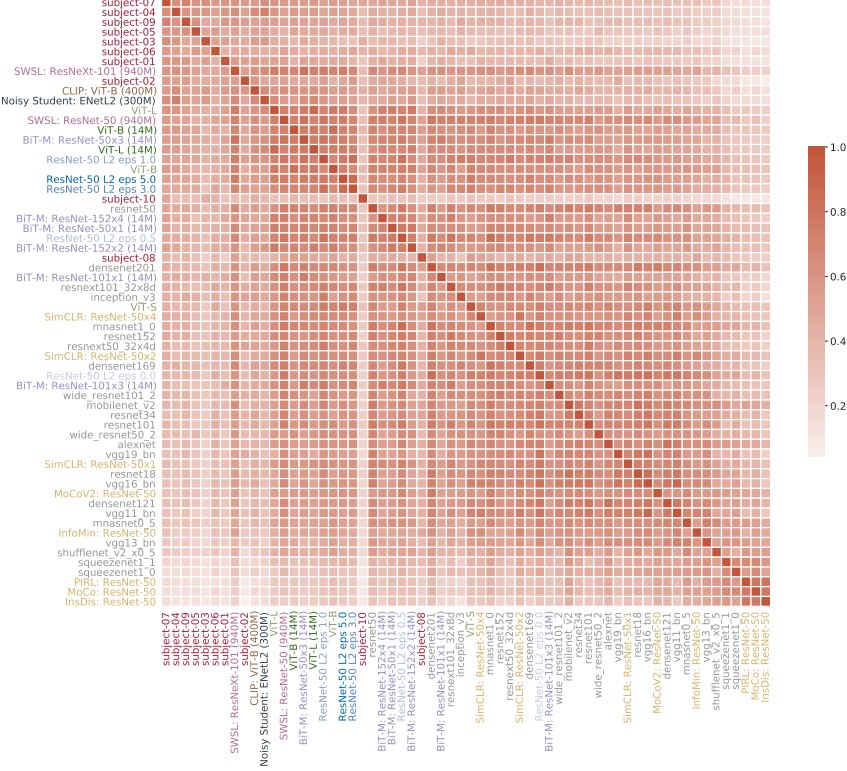

Figure 13: Error consistency for 'silhouette' images (sorted w.r.t. mean error consistency with humans).

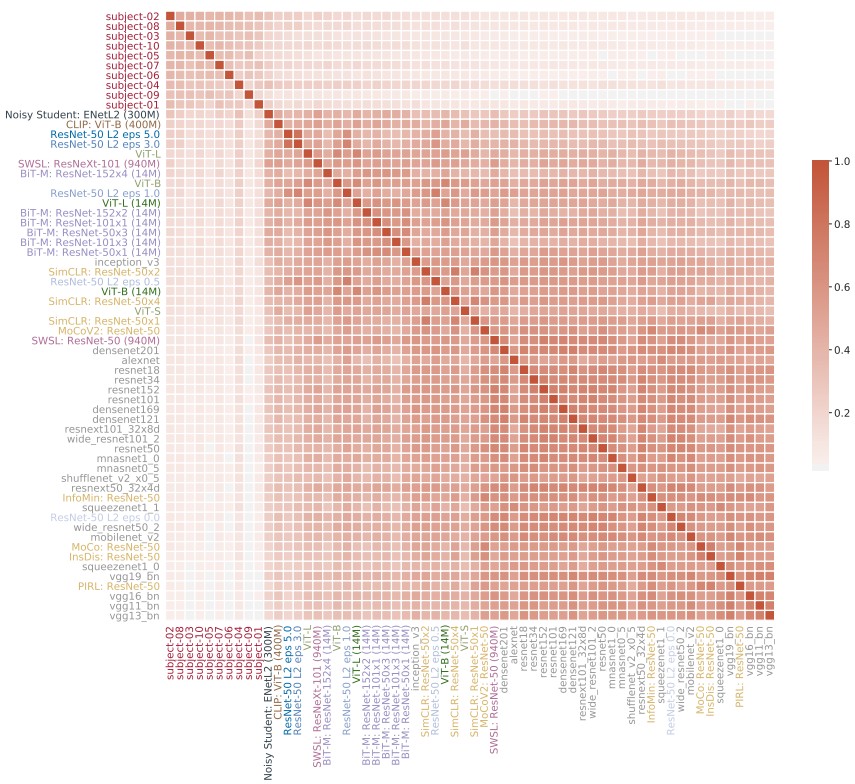

Figure 14: Error consistency for 'cue conflict' images (sorted w.r.t. mean error consistency with humans).

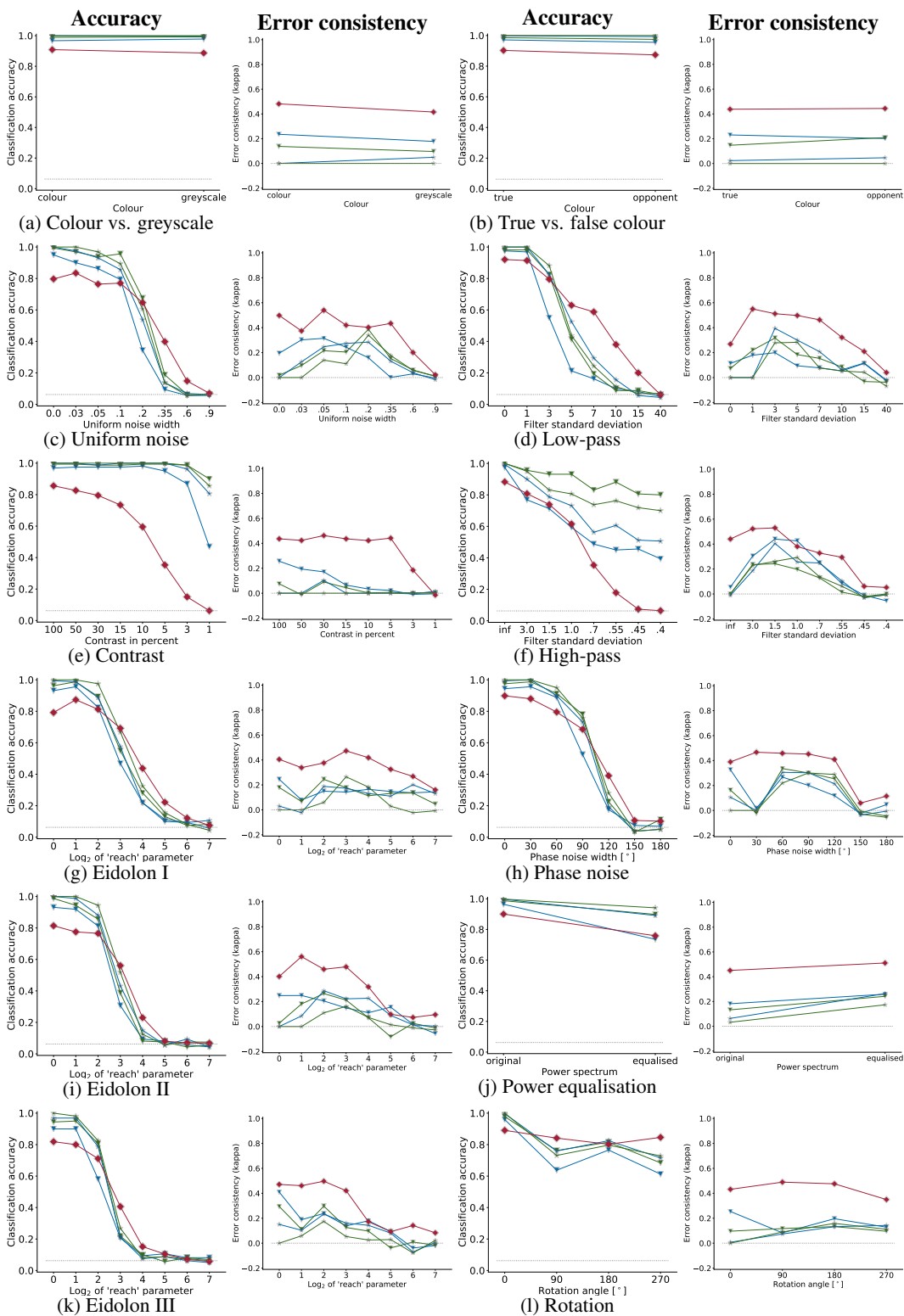

Figure 15: Comparison of self-supervised SimCLR models with supervised, augmentation-matched baseline models. Note that for better visibility, the colours and symbols deviate from previous plots. Plotting symbols: triangles for self-supervised models, stars for supervised baselines. Two different model-baseline pairs are plotted; they differ in the model width: blue models have 1x ResNet width, green models have 4x ResNet width [43]. For context, human observers are plotted as red diamonds. Baseline models kindly provided by Hermann et al. [62].

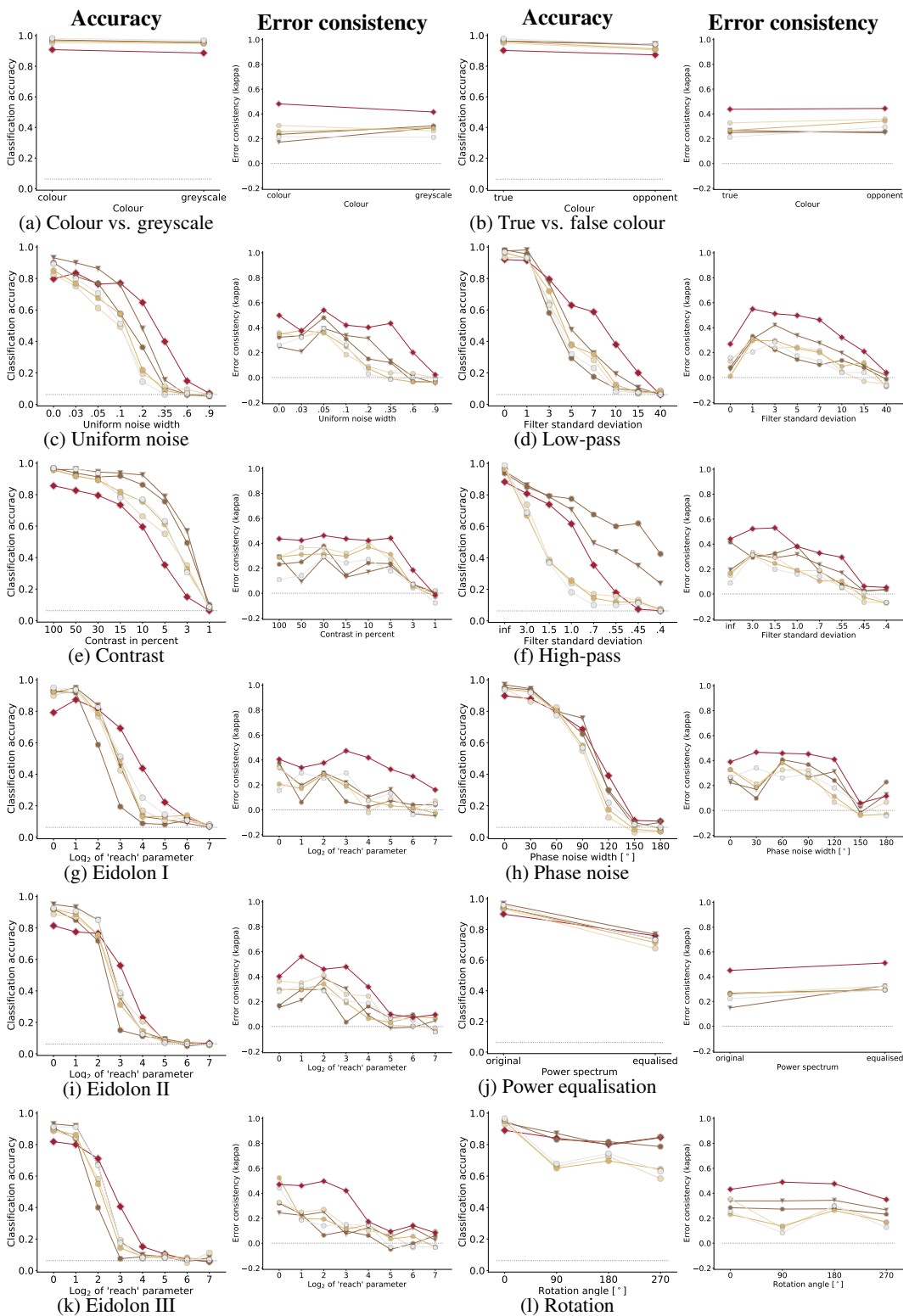

Figure 16: Do CLIP-provided labels lead to better performance? Comparison of standard ResNet-50 (light grey circles), CLIP with vision transformer backend (brown triangles), CLIP with ResNet-50 backend (brown circles), and standard ResNet-50 with hard labels (bright yellow circles) vs. soft labels (dark yellow circles) provided by evaluating standard CLIP on ImageNet; as well as humans (red diamonds) for comparison. Symbols indicate architecture type (○ convolutional, ▽ vision transformer, ◇ human); best viewed on screen. With the exception of high-pass filtered images, standard CLIP training with a ResNet-50 backbone performs fairly poorly.