# OpenReview forum: "Partial success in closing the gap between human and machine vision"
_NeurIPS.cc/2021/Conference — NeurIPS 2021 Oral_

### Official Review · Reviewer_AEmo · 2021-07-15

**Rating:** 7
**Confidence:** 5

**Summary:**

The authors employ a large-scale comparison between human classification behavior and a diverse range of machine vision models on 17 out-of-distribution (OOD) datasets as a way of attempting to comprehensively measure whether the field is making progress in closing the human-machine gap in this domain. They find that particular architectures and training objectives, as well as larger training datasets appear to be increasingly closing the gap when measuring OOD accuracy overall, but less so at the level of error patterns for individual images. They further identify subsets of their OOD task set where the human-machine gap is narrowing versus those where it is not.

**Limitations And Societal Impact:**

The authors acknowledge many limitations and there doesn't appear to be much potential for negative social impact.

**Main Review:**

While many smaller experiments have previously compared humans to CNNs, the current work is one of the most ambitious and comprehensive comparisons to date, including many architectures that have received far less study in the cognitive sciences. The conclusions are interesting, and the task suite is said to be planned for public release, which would be a very useful tool for the community.

Given the breadth of the work, it's hard to complain about many of the details. My only real concern was that the conclusions may be severely limited by the behavioral dataset used. That is, the authors used a single experimental paradigm wherein a very small set of laboratory subjects provide a larger number of judgements for a much smaller range of classes than the models which are points of comparison. There are existing behavioral datasets that are larger and pool information across many more annotators to assess image-level error patterns without having to alter the categorization task (e.g., CIFAR-10H). Other datasets focus on much richer OOD problems (ObjectNet, Omniglot) that may elicit very different human-machine deviations. For these reasons, I would have gladly traded some of the depth in model exploration in the paper for additional depth in behavioral comparison.

In the end, I have little doubt that the work represents an important analysis of the field, but I don't agree that it provides the so-called "missing human baseline", nor that datasets employed allow one to draw conclusions about the current human-machine vision gap beyond the sort of image augmentations explored by the authors.

**Time Spent Reviewing:**

3

---

> ### Author Response · Authors · 2021-08-09
> **Author response to reviewer AEmo**
>
> Dear Reviewer AEmo,\
> Thank you for reviewing our paper! We’re happy to read that you think of this work as “one of the most ambitious and comprehensive comparisons to date”, and “a very useful tool for the community”. Here’s a point-by-point response to your comments.
>
> _The authors used a single experimental paradigm wherein a very small set of laboratory subjects provide a larger number of judgements for a much smaller range of classes than the models which are points of comparison. There are existing behavioral datasets that are larger and pool information across many more annotators to assess image-level error patterns without having to alter the categorization task (e.g., CIFAR-10H)._
>
> This comment voices two potential concerns: using 16 categories instead of 1,000 (“altering the categorization task”), and using laboratory data instead of pooling information across many annotators. Regarding the first point: Indeed, for CIFAR-10 one can simply ask humans to categorize objects into those 10 classes. Naturally, this is not feasible anymore if we are to compare humans with ImageNet-trained models (1,000 classes). We therefore used a principled mapping from ImageNet classes to those 16 classes, and we now describe and motivate this choice much better in the paper, as pointed out in our response to reviewer m3s7 (second point, “Discuss and justify the choice of mapping behavioural output”) here: https://openreview.net/forum?id=QkljT4mrfs&noteId=xD3F5gjs-5V
>
> Regarding your second point (using laboratory data instead of pooling information across many more annotators), we actually think of this as an advantage rather than a limitation. In the context of  assessing image-level consistency between models and individual observers (as we do e.g. in Figure 4), pooling information across annotators is neither possible nor desirable. Furthermore, the highly controlled conditions of our psychophysical laboratory come with many advantages over crowdsourced data collection: precise timing control (down to the millisecond), carefully calibrated monitors (especially important for e.g. low-contrast stimuli), controlled viewing distance (important for foveal presentation), full visual acuity (we performed an acuity test with every observer prior to the experiment), observer attention (e.g. no multitasking or children running around during an experiment, which may happen in a crowdsourcing study), just to name a few. Jointly, these factors ensure high-quality data that will hopefully be a valuable resource for future studies.
>
> _Other datasets (e.g. ObjectNet, Omniglot) may elicit very different human-machine deviations. / The conclusions should better reflect that the results measure the human-machine vision gap for image augmentations._ \
> Agreed, and we certainly don’t mean to imply that our comparisons/datasets are the only feasible option! In fact, reviewer 6Nw6 made a similar point, and mentioned a number of other datasets. In the light of both of your points, we have addressed this important issue by:
> - re-phrasing statements to make clear we’re referring to a certain set of OOD datasets which broadly fall into the category of image distortions;
> - adding a section on related OOD datasets, in which we point out that there are many viable and important options to measure the human-machine vision gap (including ObjectNet, Omniglot, CIFAR-10H). For a longer description of the changes, please allow us to refer to our answer to reviewer 6Nw6 here: https://openreview.net/forum?id=QkljT4mrfs&noteId=JpQiXY336zn (after _"Main concern: conclusion regarding ..."_)
>
> _I don't agree that it provides the so-called "missing human baseline"._ \
> For 14 out of 17 datasets used here, no consistency-level human data was available prior to our study (e.g. Geirhos et al. 2018 showed different images to different observers, which means that we cannot use their data to assess image-level consistency). Nonetheless, we agree that the expression “missing human baseline” isn’t optimal, e.g. “baseline” is usually associated with accuracy not consistency, and might thus evoke the wrong impression. We have removed the “missing human baseline” statement to improve clarity, e.g. the abstract previously stated “To answer this question, we tested human observers on a broad range of out-of-distribution (OOD) datasets, adding the ‘missing human baseline’ by recording 85,120 psychophysical trials across 90 participants.” and now simply reads “To answer this question, we tested human observers on a broad range of out-of-distribution (OOD) datasets, recording 85,120 psychophysical trials across 90 participants.”
>
> Thanks again for your valuable suggestions, which we believe improve the clarity of our paper!

---

> > ### Comment · Reviewer_AEmo · 2021-08-31
> > **final score**
> >
> > The authors were sensitive to my comments and I am still in favor of acceptance. My score remains the same.

---

### Official Review · Reviewer_6Nw6 · 2021-07-18

**Rating:** 8
**Confidence:** 4

**Summary:**

This paper compares human and model performance on a variety of OOD benchmarks (e.g., texture-swapped images, blurred images, silhouettes, sketches) and for a variety of the most recent models (eg., CLIP, ViT, BiT, SimCLR) and shows that the best models are now on par with humans on these tasks. The paper also shows interesting remaining differences between humans and models in terms of the errors made (humans make different errors than models) and the texture bias (much stronger in models than humans).

**Ethical Concerns:**

No.

**Limitations And Societal Impact:**

Yes.

**Main Review:**

*Summary of the review:*
I recommend acceptance of this paper for its important and non-trivial conclusions, and for its impressive thoroughness (52 recent models tested on 17 OOD datasets, 90 human observers, 85,120 psychophysical trials). However, I believe that the conclusion that 'the longstanding robustness gap between humans and CNNs is closing, with the best models now matching or exceeding human performance on most OOD datasets' is too broad in scope and not currently supported by the results. I would feel comfortable increasing my rating if this claim was re-worded to more adequately reflect the scope of the results (see Main Concern).

*Strengths*
- The study is very thorough (52 recent models, 17 OOD datasets, 90 human observers, 85,120 psychophysical trials) and its main conclusions are non-trivial and have far-reaching implications for the field of computer vision.
- I found the article very well and clearly written, and pleasant to read.

*Main Concern*

I believe that the main conclusion of the study, as currently stated in the abstract  ('The longstanding robustness gap between humans and CNNs is closing, with the best models now matching or exceeding human performance on most OOD datasets'), lacks support for two reasons:
- There isn't a clear consensus on what constitutes relevant OOD benchmarks for images. The OOD tasks chosen in this paper follow a somewhat narrow definition of OOD, with the distortions only affecting pixel-level details of the images and object textures, as opposed to the overall structure of the image. For example, it would be important to see if models are now robust to unusual poses of objects in images (Alcorn et al.). See references below for more OOD datasets which affect the global structure of images and other properties.
- Humans were presented images for a short 200ms followed by a backward mask and in a 3x3 visual angle frame. I understand the motivation to remove recurrent processing, but given this potentially strong impairment to human vision, I believe that the claim that 'best models now matching or exceeding human performance' currently lacks support.

*Suggestions to improve clarity*
- Is OOD a well-defined notion for networks trained on proprietary datasets such as CLIP? For example, are we sure that the distortions presented in the OOD datasets were not present in the training set? What is the exact definition of OOD used here?
- How to reconcile the strong difference in shape/texture bias between human and models in fig3, with the overall performance match in fig1? Is the stylized dataset a particularly difficult dataset for the models tested?
- 'Observed consistency' seems to be a less informative quantity than 'Error consistency' because the overall accuracy factors in 'Observed consistency'. Why not just report 'Error consistency'?

*Other suggestions*
- Error consistency: it would be useful to have a qualitative interpretation and illustrations of the images that confuse networks and not humans and vice-versa.
- Figure 3 apparently shows a drastic failure of models to capture shape information, but would a simple linear readout trained on top of the representations of the last layer change these results? Maybe the shape information is readily available at the level of these representations but the network chose to ignore it to focus on texture. Conversely, humans naturally pay attention to shape but could probably switch to a texture-bias if asked to focus on texture.

*References*

Strike (with) a Pose: Neural Networks Are Easily Fooled by Strange Poses of Familiar Objects
Michael A. Alcorn, Qi Li, Zhitao Gong, Chengfei Wang, Long Mai, Wei-Shinn Ku, Anh Nguyen
https://arxiv.org/abs/1811.11553

Exposing Previously Undetectable Faults in Deep Neural Networks
Isaac Dunn, Hadrien Pouget, Daniel Kroening, Tom Melham
https://arxiv.org/abs/2106.00576

Small in-distribution changes in 3D perspective and lighting fool both CNNs and Transformers
Spandan Madan, Tomotake Sasaki, Tzu-Mao Li, Xavier Boix, Hanspeter Pfister
https://arxiv.org/abs/2106.16198

Natural Adversarial Examples
Dan Hendrycks, Kevin Zhao, Steven Basart, Jacob Steinhardt, Dawn Song
https://arxiv.org/abs/1907.07174

Towards Non-I.I.D. Image Classification: A Dataset and Baselines
Yue He, Zheyan Shen, Peng Cui
https://arxiv.org/abs/1906.02899
http://nico.thumedialab.com/



**Time Spent Reviewing:**

3

---

> ### Author Response · Authors · 2021-08-09
> **Author response to reviewer 6Nw6**
>
> Dear Reviewer 6Nw6,\
> Thank you very much for your thorough review! We appreciate your assessment of our work as “impressively thorough” and having “far-reaching implications for the field of computer vision”. Here’s a point-by-point response to your concerns and suggestions:
>
> _Main concern: conclusion regarding “the best models are now matching or exceeding human performance on most OOD datasets” is overly broad_ \
> Thanks for pointing this out! While we have tested a large number of models on a large number of OOD datasets (in reviewer AEmo’s words, “one of the most ambitious and comprehensive comparisons to date”), you’re absolutely right: our OOD datasets broadly fall into the category of image distortions (e.g. changes in colour, texture/style, contrast, noise level, spatial frequency content, …), and there are many other variations that make image recognition harder for CNNs such as changes in background/context, camera parameters/pose, or natural adversarial examples (just to name a few). We have taken the following steps to reflect this better in our writing:
>
> (1.) Throughout the abstract and paper, we re-phrased statements as follows. Instead of writing “the longstanding robustness gap between humans and CNNs is closing”, we now explicitly state that the “longstanding distortion robustness gap is closing”. Instead of writing how models outperform humans on “on most OOD datasets”, we now write “on most of the investigated OOD datasets”.
>
> (2.) We have added a section on “related OOD datasets” where we motivate the focus of our benchmark, and explain how our OOD datasets relate to other OOD datasets and comparisons, including those that you listed in your review. We fully agree that many other comparison datasets are relevant as well to obtain a more complete understanding of human vs. CNN OOD performance, thus the readership of our paper might benefit from a paragraph dedicated to understanding how our work fits in the larger context of OOD datasets, of which there are many interesting ones that we now explicitly cite/link to in a dedicated paragraph.
>
> (3.) Regarding the potentially better performance of humans given more time, we believe it is important to compare feedforward models to feedforward human perception, which happens within 200ms. Nonetheless, we agree that it is important to make this clear, thus we have sharpened descriptions throughout the abstract and paper. We now explicitly state that we’re referring to “feedforward human performance” or “human core object recognition performance (within 200ms)” rather than plainly writing “human performance”. Additionally, in the discussion we now mention that human ceiling performance might be higher still given longer presentation times, which is a factor that we didn’t investigate since we were aiming at a maximally fair comparison between feedforward perception in humans and feedforward models.
>
> _Is OOD a well-defined notion for networks trained on proprietary datasets such as CLIP?_ \
> Indeed, not having access to a model’s training data is problematic and blurs the boundaries between IID and OOD datasets. We consider it unlikely that CLIP was exposed to many of the exact distortions used here (e.g. “eidolon distortions” or “cue conflict”), but it is quite likely that it has had greater exposure to some conditions, e.g. grayscale or low-contrast images. We now state this caveat in the section about CLIP.
>
> _How to reconcile the strong difference in shape/texture bias between human and models in fig3, with the overall performance match in fig1? Is the stylized dataset a particularly difficult dataset for the models tested?_ \
> Yes, the dataset with a texture-shape cue conflict is one of the more challenging datasets for models. However, for other datasets (e.g. contrast, see Figure 2) many models are much better than human observers, which also contributes to the averaged performance presented in Figure 1.
>
> _'Observed consistency' seems to be a less informative quantity than 'Error consistency' because the overall accuracy factors in 'Observed consistency'. Why not just report 'Error consistency'?_ \
> While we agree that error consistency is more informative than observed consistency in the context of the investigated models (which all have fairly similar observed consistency), each of the metrics quantifies a different property and good models will need to score well on all of them, thus we decided to include plots for all metrics.
>
> _It would be useful to have a qualitative interpretation and illustrations of the images that confuse networks and not humans and vice-versa._ \
> Interesting suggestion! Analysing “diagnostic” images, i.e. stimuli that most humans correctly classified (but few networks) and vice-versa, we observe the following (qualitative) results: images that are hard for CNNs/easy for humans often fall into a few selected categories (e.g. “clock” images for edge stimuli), while images that are hard for humans/easy for CNNs don’t appear to follow this pattern. Furthermore, there are many images that all human observers correctly identified (but few networks). In these cases, the images often resemble prototypical exemplars of their category (e.g. a centered airplane in canonical view), but the image distortion (such as conversion to silhouettes) makes these images harder for CNNs than for humans.
>
> _Figure 3 apparently shows a drastic failure of models to capture shape information, but would a simple linear readout trained on top of the representations of the last layer change these results?_ \
> According to Figure 5 of https://arxiv.org/pdf/1911.09071.pdf, both shape and texture information seem to be decodable from the representation to a certain degree. Since for real-world applications the output decisions of models matter much more than the intermediate representation, our benchmark focuses on comparing output behaviour of widely used models, but it would be an interesting avenue for future research to investigate the behavioural properties of models explicitly trained to decode shape over texture features - e.g. whether this would come with improved or impaired performance on IID and OOD datasets. (Interestingly, humans apparently are unable to switch entirely to a texture bias, even if explicitly instructed to do so, cf. Figure 10 of https://openreview.net/pdf?id=Bygh9j09KX.)
>
> Again, thank you very much for reviewing our paper and for your valuable suggestions! We would appreciate it if you could let us know whether the proposed changes, especially in terms of phrasing the conclusion (your main concern), properly address your comments and suggestions.

---

> > ### Comment · Reviewer_6Nw6 · 2021-08-09
> > **Thanks + updating score**
> >
> > The proposed changes fully address my concerns about scope and clarity, and after reading the other reviews and author responses, I am comforted in my opinion that this study constitues a valuable and comprehensive comparison of feedforward processing in humans with the latest and best models for computer vision for distorted image recognition. I am updating my score accordingly (7 => 8).

---

### Official Review · Reviewer_m3s7 · 2021-07-18

**Rating:** 6
**Confidence:** 5

**Summary:**

The authors performed a large set of behavioral experiments to measure human psychophysical performance on an image classification task using a broad range of out-of-distribution datasets. They then evaluated a pool of convolutional neural networks (CNNs) on how well they matched human judgements and observed that while there is still a big gap between human and machine vision, models trained on very large datasets are starting to close that gap. The main contribution of the study is the release of the behavioral data for benchmarking models of computer vision. EDIT: Score updated 5 -> 6


**Limitations And Societal Impact:**

The authors properly addressed some of the study's limitations. The authors do not address potential negative societal impact since these are non-existent.

**Main Review:**

The paper is very well written, clear, and addresses one of the most important problems in computer vision: the difference in robustness between human and machine vision. It does so by directly comparing the performance of models in several out-of-distribution datasets with human psychophysics. Despite this, I have some concerns

This paper follows a trend of studies comparing human behavioral performance on an object recognition task with machine vision models. Studies such as  Rajalingham et al 2018 The Journal of Neurosciences, Geirhos et al 2018 NeurIPS, Geirhos et al. 2019 ICLR and Geirhos et al 2020 NeurIPS have all compared CNN performance with humans in different datasets. In that sense this study can be seen not entirely original but more as an extension of this line of research. It is difficult to see what is exactly original in this study since most of the comparisons with human behavior described here had already been done elsewhere: there is a large overlap with datasets from Geirhos et al 2018 and Geirhos et al 2020.

Furthermore, the methodology that the authors have used is also not without issues. To compare CNN models with 1,000 classes outputs with human behavioral performance in a task with 16 object categories, the authors have used a simple fixed behavioral decoder to map these different output spaces using the WordNet hierarchy. Other researchers (e.g. Rajalingham et al 2018) have used different strategies to decode behavioral choices from the model features and it is not clear which option is better. Even if the authors do not implement other strategies for decoding behavioral outputs from the models, they should at least discuss this in the paper and justify their choice.

Finally, it is not clear in which format the datasets will be made available. While I salute the authors for providing an evaluation platform to facilitate access to the benchmarks described here, it would also be important that the authors provide the raw behavioral data as well as the stimuli used so that other researchers can implement alternative metrics for comparing models to humans.

Minor issues:
I was surprised to see the omission of a citation to Rajalingham et al 2018, as it was one of the first studies to perform a detailed comparison between models to human behavioral performance in an object recognition task.

EDIT: Score updated 5 -> 6


**Time Spent Reviewing:**

8

---

> ### Author Response · Authors · 2021-08-05
> **Author response to reviewer m3s7 (1/2)**
>
> First of all: Thanks very much for your comments! We appreciate your assessment of our work as "very well written" and addressing "one of the most important problems in computer vision".
>
> Here's a detailed response to your first two concerns / comments:
>
> _Originality: "large overlap with datasets from Geirhos et al 2018 and Geirhos et al 2020"_ \
> Indeed, a number of previous studies have compared human and machine performance (e.g. Geirhos et al. 2018, 2020), finding that vanilla ImageNet-trained CNNs are a poor fit for human behavioural data, especially in terms of image-level consistency. Hence, it is an open question how the more recent generation of models performs, and which factors might contribute to improved alignment with human perception. To answer this question confidently, we first dramatically expanded the number of OOD datasets on which image-level human-machine consistency can be measured from 3 to 17 datasets, with a combined number of 20K images (instead of 1,5K as in Geirhos et al. 2020). To this end we used existing images for which we collected human data under highly controlled laboratory conditions; our results are now based on over 85K psychophysical trials (Geirhos et al. 2018 showed different images to different observers, which means that we cannot use their data to assess image-level consistency). We then performed a rigorous quantitative evaluation along the three axes _objective function_ (self-supervised, adversarially trained, CLIP language-image training), _architecture_ (e.g. vision transformers), and _dataset size_ (ranging from 1M to 1B). In contrast to previous work, we did identify models for which the match with human behavioural data is drastically improved, both in terms of generalisation performance and in terms of image-level error consistency. All these contributions are, to the best of our knowledge, clear and novel differentiations from previous work. In response to your comment, we have now sharpened our description of the contributions in the paper.
>
> _Discuss and justify the choice of mapping behavioural output, e.g. Rajalingham et al 2018 have used different choices_ \
> The behavioral performance metrics from Rajalingham et al. 2018 were not directly applicable to our analysis question since they aggregate responses over participants, whereas we were interested in measuring the consistency between models and individual observers (as shown, for instance, in Figure 4). Nonetheless, we agree that our methodology is not without alternatives, and we welcome the suggestion to discuss and motivate our choice more thoroughly, which we now do in the paper itself as follows.\
> Comparing model and human classification decisions comes with a challenge: we simply cannot ask human observers to classify objects into 1,000 classes (as for standard ImageNet models). Even if this were feasible in terms of experimental time constraints, most humans don’t routinely know the names of 120 different dog breeds. What they do know, however, is how to tell dogs apart from cats and from airplanes, chairs and boats. Those are so-called “basic” or “entry-level” categories (Rosch 1999). In line with previous work, we therefore used a set of 16 basic categories in our experiments. For ImageNet-trained models, to obtain a choice from the same 16 categories, the 1,000 class decision vector was mapped to those 16 classes using the WordNet hierarchy. Those 16 categories were chosen to reflect a large chunk of ImageNet (227 classes, i.e. roughly a quarter of all ImageNet categories is represented by those 16 basic categories). In order to obtain classification decisions from ImageNet-trained models for those 16 categories, at least two choices are conceivable: re-training the final classification layer or using a principled mapping. Since any training involves making a number of choices (hyperparameters, optimizer, dataset, ...) that may potentially influence and in the worst case even bias the results (e.g. for ShuffleNet, more than half of the model’s parameters are contained in the final classification layer!), we decided against training and for a principled mapping by calculating the probability of a coarse class as the average of the probabilities of the corresponding fine-grained classes. Why is this mapping principled? As derived by Geirhos et al. 2018 (pages 22 and 23 in the Appendix of https://arxiv.org/pdf/1808.08750.pdf), this is the optimal way to map (i.e. aggregate) probabilities from many fine-grained classes to a few coarse classes. Essentially, the aggregation can be derived by calculating the posterior distribution $p(c|x)$ of a discriminatively trained CNN under a new prior chosen at test time (here: $\frac{1}{16}$ over coarse classes); resulting in decision $C|x = argmax_C \sum\limits_{c\in C}\frac{1}{|C|}p(c|x)$.

---

> > ### Author Response · Authors · 2021-08-05
> > **Author response to reviewer m3s7 (2/2)**
> >
> > Regarding your last two concerns / comments:
> >
> > _it is not clear in which format the datasets will be made available_ \
> > We fully agree that providing the raw behavioral data as well as the stimuli is very important to enable others to use our data / stimuli for their own experiments and analyses. Short answer: everything (code/data/stimuli/models/) is available. In order to preserve anonymity we cannot link the github repository at this stage, but here’s more information on how access is provided: The toolbox and benchmark are hosted as a public github repository, which includes model evaluation, plotting code and the model zoo. The full behavioural data is directly included on github (1 csv file per observer and experiment) as part of the analysis / comparison pipeline. The full set of stimuli is made available through a github release, which has the advantage that the datasets will not slow down the process of cloning the repository. Instead, they are downloaded by the evaluation code automatically whenever a dataset is requested for the first time by a user (from the stable & permanent link provided by the github release). On average, the datasets have a size of around 50MB per dataset (minimum: 488KB for “edge” stimuli, maximum: 130MB for “cue conflict” stimuli). Overall, this achieves not only easy access by standard users, but also enables advanced users to define their own metrics or use the data/stimuli for other experiments and analyses.
> >
> > _omission of a citation to Rajalingham et al 2018_ \
> > Good catch! The Rajalingham study is indeed a widely known behavioural comparison and very relevant related work, we now cite it prominently.
> >
> >
> > We would appreciate it if you could let us know whether we have adequately addressed your concerns and comments, and thanks again for your suggestions!

---

> > > ### Comment · Reviewer_m3s7 · 2021-08-22
> > > **Concerns not completely resolved**
> > >
> > > Thank you for the responses and clarifications to my comments. Unfortunately, I still maintain a couple of concerns related to issues that I raised.
> > >
> > > In terms of the overlap with prior literature and the data sources, it is clear now to which extent the data in this paper overlaps with previous studies. While the number of different images evaluated is undoubtedly impressive, I am skeptical about the "quality" of these datasets for measuring model to human similarity. First, it is not obvious in the paper how many subjects were included in each experiment. A more distracted reader may conclude that all 90 subjects performed all the OOD datasets. *This is not correct*. Considering the combined number of images of 20k and the 85k psychophysical trials, we get around 4 subjects per condition. This is particularly relevant since each image was presented only once per subject, rendering individual subjects as very noisy estimates of error consistency. The number of subjects per condition is the only relevant number to appear in the main text and not the total number of subjects used in the experiment which is misleading and may lead to confusion! The second potential problem with the psychophysical experiments is the chosen image size of 3x3 deg. Humans perform object recognition over larger spatial extents, and therefore this data may underestimate the true generalization abilities of humans in these OOD datasets.
> > >
> > > Regarding the methods used to evaluate models and compare their outputs with human choices, the approach followed by the authors is not without problems. For example, since the human psychophysical experiments were done with stimuli spanning 3deg, directly presenting these images to the models and mapping their ImageNet labels to these 16 categories, assumes that the models' input extent is 3deg. This is not in agreement in multiple studies using ANNs as models of primate vision (Schrimpf, Kubilius et al 2018, Cadena et al 2019, Marques et al 2021), which have shown that these models better approximate primate vision when processing images between 6 to 8deg. Beyond this, training a behavioral decoder from the model features to the classes and particular image distributions in the study is, in my opinion, a better approach for evaluating whether a model is more aligned with human vision. The last layer of a ImageNet trained model is optimized for the 1000 classes and the image distribution of that dataset. It is well known that those outputs will map poorly to other datasets that change some aspects of the input images. What is interesting from the perspective of human-to-model comparisons, for me, is whether the model learns representations that can be linearly read and mapped to human behavioral choices. While I understand the technical difficulties of following this approach, this for me, is the proper way to compare models to humans. What the authors test on this study is whether off-the-shelf models perform well in OOD datasets without fine-tuning, which is an interesting question of its own, but a very different one than comparing models to humans. In that sense, this study is very similar to Taori et al NeurIPS 2020, which is also surprisingly not cited, and evaluated robustness to natural distribution shifts including the following conclusion on the abstract: "Moreover, most current techniques provide no robustness to the natural distribution shifts in our testbed. The main exception is training on larger and more diverse datasets, which in multiple cases increases robustness, but is still far from closing the performance gaps."

---

> > > > ### Author Response · Authors · 2021-08-27
> > > > **Author response to remaining / new concerns**
> > > >
> > > > Dear Reviewer m3s7,\
> > > > Thanks for your response. We’re glad we were able to clarify a number of questions. Here’s a response to your remaining concerns (some of them were, of course, not addressed by our initial response since they hadn’t been raised in the initial review):
> > > >
> > > > _(1.) “it is not obvious in the paper how many subjects were included in each experiment.”_ \
> > > > Actually, the number of subjects and trials for each experiment are listed in Figures 7 and 8 (referenced in line 73, i.e. the beginning of the methods section). We’re happy to rephrase this line as “... Figures 7 and 8 in the Appendix, which also state the number of subjects and trials per experiment” to make the connection more apparent. Our approach of investigating few observers in a high-quality laboratory setting performing many trials is known as the so-called “small-N design”, the bread-and-butter approach in high-quality psychophysics (see, e.g., the recent review of Smith & Little 2018, “Small is beautiful: In defense of the small-N design”, https://link.springer.com/article/10.3758/s13423-018-1451-8). This is in contrast to the “crowdsourcing approach” (many observers in a noisy setting performing fewer trials each).
> > > >
> > > > _(2.) “The second potential problem with the psychophysical experiments is the chosen image size of 3x3 deg. Humans perform object recognition over larger spatial extents, and therefore this data may underestimate the true generalization abilities of humans in these OOD datasets.”_ \
> > > > We’re not sure we share the intuition that human object recognition accuracies should be higher for stimuli that are presented in regions with lower visual acuity. We opted for central foveal presentation of our stimuli precisely for the reason that this ensures high visual acuity; furthermore, if 3x3 degrees were too small then human observers would also have problems recognizing undistorted images in our setup, which is not the case.
> > > >
> > > > _(3.) “Training a behavioral decoder from the model features to the classes and particular image distributions in the study is, in my opinion, a better approach for evaluating whether a model is more aligned with human vision. What is interesting from the perspective of human-to-model comparisons, for me, is whether the model learns representations that can be linearly read and mapped to human behavioral choices. What the authors test on this study is whether off-the-shelf models perform well in OOD datasets without fine-tuning, which is an interesting question of its own, but a very different one than comparing models to humans. In that sense, this study is very similar to Taori et al NeurIPS 2020”_ \
> > > > “One operational definition of ‘understanding’ object recognition is the ability to construct an artificial system that performs as well as our own visual system” (DiCarlo, Zoccolan & Rust 2012, “How Does the Brain Solve Visual Object Recognition?”, https://www.sciencedirect.com/science/article/pii/S089662731200092X). In this sense, asking the question of how the latest generation of computer vision models performs in comparison to human observers on exactly the same images, without any fine-tuning, decoding or architectural modifications (as we do in our study), is highly relevant for the community interested in comparing models to humans. We agree that other choices are interesting as well, such as asking whether models can be trained / decoded to match human responses, but this is not the scope of our paper. (Thanks for the suggestion to link our findings to Taori, which we now do; although we would like to point out that the Taori study did not compare models to humans, which is a central aim of our paper).

---

> > > > > ### Comment · Reviewer_m3s7 · 2021-09-01
> > > > > **Score update**
> > > > >
> > > > > My major concern with the psychophysical data is not only that few subjects were used but that in conjunction with the fact that for each subject only one trial per image was presented. While it is true that the number of subjects per experiment is mentioned in the caption of Figures 7 and 8 in the supplementary material, this information should be made more obvious in the main text. Instead, it is mentioned that 90 subjects were used in the experiments which tells us nothing about the statistical power of the data (each experiment only had on average around 4 subjects).
> > > > >
> > > > > The problem with the quote from DiCarlo, Zoccolan & Rust 2012 in this context is that the artificial systems explored in this study were not built as models of object recognition in the brain. While it is true that one can use these CNNs as models of the primate ventral stream and object recognition behavior, this is done either by mapping features in the models to neurons in the brain or by decoding behavioral choices from the models’ features on the later layers. Not by taking their outputs on datasets that these models were not designed to solve and comparing them directly with human behavioral choices. This, to me, seems like a straw-man.
> > > > >
> > > > > While I strongly disagree with the authors in these two points (importance of higher number of repetitions per trial and/or subjects, and relevance of directly comparing the outputs of the models with human behavioral choices), I am aware that these boil down to personal preference. Giving the benefit of the doubt and taking into consideration the extent of the analyses in this study and its importance to the field, I am updating the score to 6.

---

> > > > > > ### Author Response · Authors · 2021-09-01
> > > > > > **Thanks & last clarifications**
> > > > > >
> > > > > > Dear Reviewer m3s7,
> > > > > >
> > > > > > Thanks for your response for taking the time to engage in a discussion.
> > > > > >
> > > > > > As you pointed out, we appear to have different opinions regarding the value of a small-N-design and the value of testing models without re-training/decoding. We will take this as an opportunity to justify our methodological choices in the main paper, and to explain why we made these choices; while at the same time mentioning that these choices are not without alternatives (as e.g. explained by you in your review).
> > > > > >
> > > > > > One last comment regarding your concern about the relationship between the number of subjects and the statistical power of the data: In a small-N-design (as in https://link.springer.com/article/10.3758/s13423-018-1451-8), statistical power comes from collecting many trials per subject, which is in contrast to e.g. crowdsourcing designs where many subjects perform fewer trials. For instance, if a single observer correctly identifies 1,000 out of 1,280 images (1,280 images per observer were used in many of our experiments), then the proportion correct is 78.13% +/- 2.26%, where the +/- 2.26% correspond to a 95% binomial confidence interval - which is already very small for just a single observer! To quote the article linked above, *"One recommendation to remedy the replication crisis is to collect larger samples of participants. We argue that this recommendation misses a critical point, which is that increasing sample size will not remedy psychology’s lack of strong measurement, lack of strong theories and models, and lack of effective experimental control over error variance. In contrast, there is a long history of research in psychology employing small-N designs that treats the individual participant as the replication unit, which addresses each of these failings, and which produces results that are robust and readily replicated."* In this sense, the statistical power in our experiments comes from the large number of trials per subject. (Showing the same image twice to a single observer is not a feasible option, since this would invoke memory effects which would confound the findings).
> > > > > >
> > > > > > In any case, thanks again for the discussion, which brought up a number of important points that we will be happy to incorporate in the paper!

---

> > > > > > > ### Comment · Reviewer_m3s7 · 2021-09-01
> > > > > > > **Answer is still missing the fact that only one trial per image is shown to each subject**
> > > > > > >
> > > > > > > Regarding the example given in the authors' reply:
> > > > > > > "For instance, if a single observer correctly identifies 1,000 out of 1,280 images (1,280 images per observer were used in many of our experiments), then the proportion correct is 78.13% +/- 2.26%, where the +/- 2.26% correspond to a 95% binomial confidence interval - which is already very small for just a single observer!"
> > > > > > >
> > > > > > > While it is true that presenting many different images per subject gives a very good estimate of the overall accuracy on the corresponding dataset with a small confidence interval, for each image there is a single sample to evaluate the subject's accuracy on that image. *This is where my main concern lies since the model to human comparison measures the alignment of these at the image-level (on an image by image basis)!* Even in a controlled condition, psychophysics measurements have some experimental noise associated to them and by having a single trial at the level of analysis (individual image) is problematic.

---

> > > > > > > > ### Author Response · Authors · 2021-09-01
> > > > > > > > **Error consistency is based on all trials of an experiment (not just a single image)**
> > > > > > > >
> > > > > > > > "For each image there is a single sample to evaluate the subject's accuracy on that image": this would be problematic if we were to make statements about a single observer's accuracy on a single image. Fortunately, error consistency doesn't require this. Instead, error consistency compares two observers image-by-image on *all* trials of an experiment. For instance, a single error consistency score obtained from comparing two observers, in the example above, would be based on 2 x 1,280 trials (not on a single image).
> > > > > > > >
> > > > > > > > We fully agree that humans - even in highly controlled lab conditions - are to a certain degree "noisy" (as you pointed out). This is why we measure and report human-to-human consistency for every experiment, and use it as a reference value when assessing model-to-human error consistency (see e.g. Figure 1d, Figure 4, Figure 5ab). We cannot expect models to be more consistent with a human observer than one observer with another observer, but we can compare human-to-model consistency against human-to-human consistency even if human-to-human consistency is less than perfect due to e.g. internal noise or inter-individual differences.
> > > > > > > >
> > > > > > > > Does this address your concern?

---

### Official Review · Reviewer_bjmw · 2021-07-21

**Rating:** 9
**Confidence:** 4

**Summary:**

This paper addresses a very interesting and important question: to what extent the recent improvements in object recognition models make them closer to human perception. The authors do this by collecting human data on a wide range of ood benchmarks modeled after ImageNet, evaluating the behavior of a large number of recently developed ImageNet models on these benchmarks, and comparing them with the human responses. The authors also open-source all their data and analysis tools so people can evaluate their own models on the same benchmarks and compare them with human behavior in a fine-grained manner (on an image-by-image basis).

**Limitations And Societal Impact:**

I think the authors do discuss the limitations of this work adequately in their discussion section. For me, the most important limitation is that static object recognition is an important, but limited facet of human perception. Extending this kind of detailed analysis to a broad range of other human perceptual skills in the future would be important.


**Main Review:**

I think this is overall a solid paper packed with important and, to my knowledge, novel contributions and insights:

1) The human data itself and the open-source tools made available for evaluating and analyzing new models are very useful contributions.

2) The authors show that several recent models make significant progress in closing the gap with human behavior. This seems to be true even with respect to the most stringent comparison metric: i.e. image-by-image error consistency with humans. The most important contributor to this progress seems to be pretraining on large datasets (e.g. CLIP, SWSL, BiT).

3) Self-supervised models do not perform any differently from their supervised counterparts when differences in data augmentation pipelines are accounted for.

4) Transformer models in general appear to be better than comparable convnets both in terms of ood performance and in terms of consistency with human observers.

Perhaps, one small suggestion for improvement I can make is some kind of factorial quantitative analysis of the effects of different factors considered here both on ood performance and on consistency with human observers. Currently the authors are making these judgements qualitatively by looking at some plots and concluding, for example, “models pre-trained on large datasets perform better on ood benchmarks and are more consistent with human observers” or “transformers perform better than convnets” etc., which is fine, but I think a more quantitative, factorial analysis would be even better, because, among other things, it would, for example, also potentially give us more information about the interactions and comparative effect sizes.


**Time Spent Reviewing:**

3

---

> ### Author Response · Authors · 2021-08-09
> **Author response to reviewer bjmw**
>
> Dear Reviewer bjmw,\
> Thank you for reviewing our paper! We’re glad to hear that you found our work to be a “solid paper packed with important and, to [your] knowledge, novel contributions and insights”. Please find a response to your analysis suggestion below.
>
> _some kind of factorial quantitative analysis of the effects of different factors considered here both on ood performance and on consistency with human observers_ \
> We think this is an interesting & valuable suggestion! Admittedly, we weren’t 100% sure about the type of analysis that would make most sense here: A standard “factor analysis” in the sense of https://en.wikipedia.org/wiki/Factor_analysis, i.e. attempting to discover latent variables in the data probably wouldn’t lead to the desired effect, since we would rather want to know the influence of known independent variables (architecture: transformers vs. ConvNets; data: small vs. large; objective: supervised vs. self-supervised) on known dependent variables (OOD performance and error consistency with humans). We therefore performed a regression analysis where we modelled the influence of those predictors on OOD performance, and on error consistency with human observers in two separate linear regression models (one per dependent variable). To this end, we used incremental model building, i.e. starting with one significant predictor and subsequently adding predictors if the reduction of degrees of freedom is justified by a significantly higher degree of explained variance (alpha level: .05). Both error consistency and accuracy, for our 52 models, followed (approximately) a normal distribution, as confirmed by density and Q-Q-plots. That being said, the fit was better for error consistency than for accuracy.
>
> The final regression model for error consistency showed:
> - a significant main effect for transformers over CNNs (p = 0.01872 *),
> - a significant main effect for large datasets over small datasets (p = 1.06e-05 ***),
> - a significant interaction between dataset size and objective function (p = 0.00957 **),
> - no significant main effect of objective function (p = 0.16909, n.s.)
>
> Residual standard error: 0.02017 on 47 degrees of freedom\
> Multiple R-squared:  0.5676,    Adjusted R-squared:  0.5308\
> F-statistic: 15.43 on 4 and 47 DF,  p-value: 3.973e-08
>
> The final regression model for OOD accuracy showed:
> - a significant main effect for large datasets over small datasets (p = 7.98e-08 ***),
> - a significant interaction between dataset size and architecture type (transformer vs. CNNs; p = 0.0385 *),
> - no significant main effect for transformers vs. CNNs (p = 0.7237, n.s.)
>
> Residual standard error: 0.05655 on 48 degrees of freedom\
> Multiple R-squared:  0.556,    Adjusted R-squared:  0.5283\
> F-statistic: 20.04 on 3 and 48 DF,  p-value: 1.462e-08
>
> Limitations: a linear regression model can only capture linear effects; furthermore, diagnostic plots showed a better fit for the error consistency model (where residuals followed the expected distribution as confirmed by a Q-Q-plot) than for the OOD accuracy model (where residuals weren’t perfectly normal distributed).
>
> That being said, we think this analysis is an interesting addition to our paper since, as you pointed out, it not only provides statistical support to our conclusions but also revealed interaction effects. Thanks again for this suggestion!

---

> > ### Comment · Reviewer_bjmw · 2021-08-27
> > **thank you**
> >
> > I appreciate that the authors took the time to carry out the analysis I suggested. I have decided to increase my score as a result. I think it would be useful to include these results in the final version of the paper. As I already indicated in my initial review, I still believe this is a very strong paper with many valuable contributions and insights.

---

> > > ### Author Response · Authors · 2021-08-27
> > > **Thanks!**
> > >
> > > Thanks for your response! We couldn't agree more: the analysis you suggested is a great addition to the paper and we will make sure to include it in the revised version (probably as a combination of reporting the full details in the Appendix and summarizing the key findings in the main paper).

---

### Author Response · Authors · 2021-09-01
**Author summary of rebuttal discussion**

We would like to thank all four reviewers for their time and valuable feedback!

We appreciate their assessment of our work as a **“very strong paper with many valuable contributions and insights”** (bmjw), **"very well written"** and addressing **"one of the most important problems in computer vision"** (m3s7), **“impressively thorough”** and having **“far-reaching implications for the field of computer vision”** (6Nw6), **“one of the most ambitious and comprehensive comparisons to date”**, and **“a very useful tool for the community”** (AEmo).

We received numerous helpful suggestions; this is a summary of main concerns and how we addressed them:

- In response to reviewer bmjw, we performed a regression analysis where we modelled the influence of certain predictors on OOD performance, and on error consistency with human observers in two separate linear regression models
- In response to reviewers 6Nw6 and AEmo, we added a section on related OOD benchmarks, and clarified the scope of our paper (measuring distortion robustness rather than measuring OOD robustness in general, which would be overly broad).
- In response to reviewers AEmo and m3s7, we improved the description of how behavioural responses are mapped, and discuss/link to a theoretical justification
- In response to reviewer m3s7, we sharpened our description of the contributions in the paper, provided details on how the toolbox will be open-sourced, and better justify our methodological choices

All reviewers have already taken the time to consider our responses and largely indicated being happy with them. We had a longer discussion with reviewer m3s7, during which we were able to clarify a number of points and identify some aspects on which we disagree (e.g. the value of using a small-N-design as in https://link.springer.com/article/10.3758/s13423-018-1451-8, which we employ in our well-controlled psychophysical laboratory and which is in contrast to e.g. a crowdsourcing study); we will take this as an opportunity to clarify and justify our choices better in the main paper.

Thanks again to all reviewers for their time, effort and constructive feedback!

---

### Decision · Program_Chairs · 2021-09-27

**Decision:**

Accept (Oral)

**Comment:**

The reviewers were enthusiastic about the breadth and quality of this work. I agree; this is an exceptional paper that will be an important resource for the community in comparing human and machine vision, especially with the open-source evaluation software.

The reviewers offered important suggestions about the wording of some of the conclusions, which the authors were receptive to in the rebuttal. The regression analysis suggested by R-bjmw is also very useful. These changes will further improve the final manuscript.